# Amination of Graphene Oxide Leads to Increased Cytotoxicity in Hepatocellular Carcinoma Cells

**DOI:** 10.3390/ijms21072427

**Published:** 2020-03-31

**Authors:** Milena Georgieva, Bela Vasileva, Giorgio Speranza, Dayong Wang, Kalin Stoyanov, Milena Draganova-Filipova, Plamen Zagorchev, Victoria Sarafian, George Miloshev, Natalia Krasteva

**Affiliations:** 1Institute of Molecular Biology “Roumen Tsanev”, Bulgarian Academy of Sciences, 1113 Sofia, Bulgaria; milenageorgy@gmail.com (M.G.); belavas@outlook.com (B.V.); karamolbiol@gmail.com (G.M.); 2Functional Materials and Photonic Structure, Center for Materials and Microsystems, Fondazione Bruno Kessler, I-38123 Povo, Italy; speranza@fbk.eu; 3Department of Biochemistry and Molecular Biology, Medical School in Southeast University, Nanjing 210009, China; dayongw@seu.edu.cn; 4Department of Automation, University of Chemical Technology and Metallurgy, 1756 Sofia, Bulgaria; kalin.stoyanov@uctm.edu; 5Department of Medical Biology, Medical Faculty, Medical University—Plovdiv, 4000 Plovdiv, Bulgaria; milena_draganovafilipova@abv.bg (M.D.-F.); sarafian@abv.bg (V.S.); 6Research Institute at Medical University-Plovdiv, Bulgaria; 15A Vassil Aprilov, blvd, 4000 Plovdiv, Bulgaria; 7Department of Physics and Biophysics, Faculty of Pharmacy, Medical University—Plovdiv, 4000 Plovdiv, Bulgaria; plamenz@gbg.bg; 8Institute of Biophysics and Biomedical Engineering, Bulgarian Academy of Sciences, 1113 Sofia, Bulgaria

**Keywords:** cytotoxicity, genotoxicity, HepG2, nanoparticle functionalization, GO, hydroxylamine, haGO-NH_2_

## Abstract

Clinically, there is an urgent need to identify new therapeutic strategies for selectively treating cancer cells. One of the directions in this research is the development of biocompatible therapeutics that selectively target cancer cells. Here, we show that novel aminated graphene oxide (haGO-NH_2_) nanoparticles demonstrate increased toxicity towards human hepatocellular cancer cells compared to pristine graphene oxide(GO). The applied novel strategy for amination leads to a decrease in the size of haGO-NH_2_ and their zeta potential, thus, assuring easier penetration through the cell membrane. After characterization of the biological activities of pristine and aminated GO, we have demonstrated strong cytotoxicity of haGO-NH_2_ toward hepatic cancer cells—HepG2 cell line, in a dose-dependent manner. We have presented evidence that the cytotoxic effects of haGO-NH_2_ on hepatic cancer cells were due to cell membrane damage, mitochondrial dysfunction and increased reactive oxygen species (ROS) production. Intrinsically, our current study provides new rationale for exploiting aminated graphene oxide as an anticancer therapeutic.

## 1. Introduction

Recently, novel therapeutic approaches, based on different nanoparticles, have been identified as a promising multi-modal approach for enhancing therapeutic efficacy and reducing side effects associated with cancer treatment [1,2,3]. Nanoparticles are associated with a more targeted localization in tumors and cellular uptake on an active mode, making it possible to achieve controlled drug-released delivery and when possible specific gene transfection [4,5].

Graphene oxide (GO) nanoparticles are broadly used nowadays due to GO properties and characteristics. GO is a two-dimensional carbon nanomaterial with unique combination of physical and chemical properties, and great potential for use as drug carrier in targeted cancer treatment [6,7]. Structurally, GO presents a graphene-like sheet, laced with carboxyl groups on its edge and hydroxyl/epoxy groups on its basal plane [8]. The layered structure has both, a large surface area and abundant oxygen-containing groups facilitating functionalization of GO with bioactive molecules to increase loading of hydrophobic or aromatic anticancer drugs and to ease cancer-cell targeting. Cancer-cell targeting antibodies and molecules, such as DNA, peptides, polymers, etc. are easily adsorbed onto the GO surface through non-covalent π–π interactions or electrostatic interactions, while the organic molecules, including drugs, are able to covalently conjugate with GO [9]. Additionally, the GO nanosheets, produced by the modified Hummer’s method, demonstrate stability in water suspensions and high biocompatibility [10], mainly because of the presence of hydrophilic oxygenated groups, which makes them ideal materials for application in drug delivery [11].

The toxicity of GO has been focused recently, but there is currently a lack of consistency in this regard. According to some authors, GO has a minor effect on the viability and morphology of different cancer cells [11,12], while others have proven that GO may generate oxidative stress, causing adverse effects on the viability of HaCaT (non-tumor skin keratinocytes) and liver cancer cells [13,14]. Moreover, GO treatment could lead to structural deformation of mitochondria, thus, affecting the mitochondrial membrane potential [15]. Many studies prove that the cellular toxicity of GO in vitro is closely related to its surface functionalization [16]. The latter provokes alterations in the physicochemical properties of GO, which impacts its application in cancer therapy. Exposure to COOH-functionalized GO nanoparticles, for example, can cause passive apoptosis in T-lymphocytes, while PEG-modified GOs cause severe hemotoxicity in the same cells by inducing membrane damage [17]. PEGylation reduces the non-specific binding of GO to biological membranes and improves its in vivo pharmacokinetics for better tumor targeting [18,19]. In our previous study, we found that exposure to ammonia-modified GO NPs could induce cell cycle arrest in the G2/M phase, thus, significantly increasing the apoptosis rate and the generation of reactive oxygen species (ROS) in colorectal Colon 26 and lung A549 cancer cells, but did not influence the viability of non-tumor embryonic stem cells [20,21]. Our findings highlight the potential of GOs NPs to be used as anticancer drugs. The data prove that functionalization of GO can be used to modulate GO cytotoxicity and genotoxicity, and to design effective strategies for cancer therapy. Importantly, the better we understand the physiochemical properties of NPs, their interactions with the cells and the possible toxicity mechanisms, the more precisely their potential biomedical applications could be outlined. Therefore, further studies are warranted to fully understand the cellular toxicological mechanism of functionalized GO NPs.

In view of the insignificant knowledge about the mechanisms of toxicity of aminated GO NPs, this study was designed to investigate the biological mechanisms of cytotoxicity of newly synthesized and functionalized GO NPs on human hepatocellular carcinoma (HepG2) cells. Hepatocellular carcinoma (HCC) is an aggressive tumor typically occurring in patients with chronic liver disease [22]. It is the second leading cause of cancer death in East Asia and sub-Saharan Africa and the sixth most common in Western countries [23,24]. The prevalence of HCC is increasing due to the increasing incidence of hepatitis infection, obesity, and metabolic syndrome, as well as increased survival of patients with liver disease [25]. Actually, the HCC is one of the biggest challenges in cancer treatment, due to its different molecular pathways, causing agents, and late diagnosis. Depending on the disease stage, there are several treatments, such as surgical resection, ablation, transplantation, chemoembolization, and medication with Sorafenib [26,27,28]. However, all treatments bring side effects that impair the quality of patients’ life. In addition, the high toxicity and relative non-specificity of conventional anticancer drugs, used for the treatment of HCC, impede long-term application [29].

We have developed an easy, one-step protocol for amination of GO by hydroxylamine, which is much simpler than the commonly used protocols for amination of GO by ammonia. We aimed to study the potential cytotoxicity of pristine and hydroxylamine aminated GO (haGO-NH_2_) on HepG2 cells, focusingon the assessment of the rate of ROS production and mitochondrial dysfunction. Our results showed that hydroxylamine aminated GO induced cytotoxicity, oxidative stress, membrane leakage and mitochondrial dysfunction in HepG2 cells while DNA damage was insignificant. To the best of our knowledge, these data are the first that link aminated GO with potential impairment of the mitochondrial metabolism and will be of profound clinical significance for the development of aminated GO NPs, either as stand-alone therapeutics, or in combination with other anticancer drugs.

## 2. Results

### 2.1. Structural and Biophysical Characterization of GO and haGO-NH_2_Nanoparticles

Graphene oxide nanoparticles, pristine and aminated by hydroxylamine were characterized structurally and biophysically by subsequently evaluating their chemical composition, morphology, size, hydrodynamic diameter and zeta potential. X-ray photoemission spectroscopy (XPS) was utilized to estimate the amination of GO upon modification of GO with hydroxylamine. XPS allows specific characterization of GO materials through both, atomic survey scans and high-resolution atomic scans. Peak fitting of high resolution XPS is a powerful and commonly used tool for detection of specific chemical modifications of different materials. Deconvolution of high resolution N1s XPS provided evidence that the nitrogen is covalently attached to GO in the haGO-NH_2_ NPs. The N1s fit components were assigned to C=N and amine bonds (398.4 eV), amide or imide bonds (399.89 eV) and protonated amines (401.6 eV). XPS survey scans proved that haGO-NH_2_ contains amino groups because haGO-NH_2_ amine peak was absent in pristine GO and the peak at higher BE is much more pronounced (Figure 1). Additionally, the high-resolution XPS C1s and O1s spectra demonstrated an alteration in the percentage of bonds in GO and haGO-NH_2_ NPs suggesting that amination affected the functional groups in GO. For example, the percentage of C–O (532.65 eV) in haGO-NH_2_ decreased with 17.3%, while the percentage of C=O (531.49 eV) increased with 18.62% when compared to GO. Also, a peak at 1022.9 was registered in GO-NH_2_ NPs corresponding to the ZnO_2_ pointing contamination of the sample.

Transmission electron microscopy (TEM) was used to observe the general morphology of the GOs NPs. The micrographs of pristine and aminated GO NPs (Figure 2A) revealed a two-dimensional sheet-like structure, consisting of one or several layers. The transparency of GO sheets suggested the formation only of few layers. The GO sheets had an irregular, wrinkled shape with sharp edges. Compared with pristine GO, haGO-NH_2_ sheets were more wrinkled (the right micrograph at Figure 2A) with no other morphological changes being detected. Similar morphologies for both, pristine and aminated by ammonia GO sheets have been observed in our previous studies [21], indicating that the different methods of amination of GO resulted in a similar effect on GO morphology, namely increasing the level of crumpling of the modified GO sheets.

Dynamic light scattering (DLS) is a particularly important technique for determining nanoparticle size and size distribution in aqueous suspensions. We have implemented DLS measurements in aqueous solutions to elucidate the size of the studied GO and haGO-NH_2_ NPs (Figure 2B). Samples were dispersed in water (at 1 mg/mL), followed by sonication for 60 min and then analyzed with Zetasizer. The pristine GO was found to have very wide size distribution ranging from 280 nm to 6.4 µm (Figure 2B) divided in two fractions: a small fraction of 19.24 % that encompasses the particles ranging from 289 nm to 818 nm in size with average particle size of 515 nm and a main fraction of 80.76% with particles in the range of 1.64 µm to 6.54 µm with average particle size of 3.6 µm. Aminated GO NPs were more homogeneous and smaller in size ranging from 102 nm to 1.944 µm with average particle size of 594 nm.

The zeta potential (ζ-potential) of GO and haGO-NH_2_ NPs was measured in order the colloidal stabilities of nanoparticles to be determined. Many data show that a colloidal dispersion is stable when a force causes the mutual repulsion of the particles [30]. In general, a particle suspension with a zeta potential of around −30 mV is considered as a stable dispersion [31]. The table on Figure 2C contains zeta potential values for both types of pristine and aminated GO samples at room temperature. The zeta potential values for pristine GO samples were −33.7 mV when dispersed in water and increased to −12.28 mV after amination. Pristine GO suspension demonstrated higher colloidal stability than aminated GO. Taken together, the results from Zetasizer measurements suggested that amination by hydroxylamine decreased the size of haGO-NH_2_ NPs and their zeta potential, suggesting easier penetration of the nanoparticles through the cell membrane.

### 2.2. Cytotoxicity of Hydroxylamine Modified GO (haGO-NH_2_) Nanoparticles Is Increased in HepG2 Cells While Cell Morphology Remains Unchanged

The biological effects of GO nanoparticles, pristine and hydroxylamine aminated, were tested on HepG2 cells. Cytotoxicity was measured after 24 h exposure of the cells to different concentrations of the two types of NPs. The tested concentrations were 4, 10, 25 and 50 μg/mL f.c. The optical density (OD) of cells treated with GO and haGO-NH_2_ was spectrophotometrically measured at 450 nm wavelength (Figure 3). To illustrate the trend by which both types of GO NPs exert their cytotoxic activity on HepG2 cells we have built linear regression models (Figure 3A,B). The green dots represent the measured values. The R squared statistics were also included. The trend towards increasing the concentrations of the GO and the measured cytotoxicity is obvious (Figure 3A), but the observed effect of the exposure to GO was not strong. Moreover, considering the measuring error (notice the rather wide variation), one can conclude that the effect of GO is almost negligible. The level denoted by “K” marks the values without GO. Our results showed that the pristine GO NPs reduced cell viability only after treatment with higher concentrations of 25 and 50 μg/mL, while the lower concentrations of 4 and 10 μg/mL had slight stimulating effect on cells, assumed as the so-called hormesis effect.

Figure 3B demonstrates the linear regression model for the effect of the hydroxylamine modified haGO-NH_2_. It is easily observed that, even with the lowest concentration (4 μg/mL) of the aminated NPs, the OD has decreased significantly down to 0.7 with variation between 0.61 and 0.76 and gradually decreased until 0.30, comparing the levels of “K” at zero concentration of haGO-NH_2_ (1.48). Also one can see that the trend is not linear but quadratic - the regression model has correlation coefficient 0.91 (Figure 3B), which demonstrates much better fit than the linear model over the same data from the previous example. This model also prompts saturation of the effect of haGO-NH_2_ at a concentration of 50 μg/mL. Observation of neutral red labeled hepatocytes did not show any significant alterations in cell morphology (Figure 3C).

Cytotoxicity was additionally assessed by measurement of the leakage of lactate dehydrogenase (LDH). LDH is an enzyme, which is released into the surrounding extracellular space following cell exposure to cytotoxic compounds. When the cell membrane integrity is compromised, the detection of LDH in the culture medium can be used as a cell-death marker [32]. To evaluate the levels of LDH leakage after incubation with both types of GO NPs, HepG2 cells were treated with GO and haGO-NH_2_ for 24 h, followed by measurement of LDH levels (Figure 4). Cells with a fully compromised cell membrane were obtained after treatment with Triton X-100. The LDH release values were quantified as a percentage of the LDH release in Triton X-100 treated cells which was taken as 100%. The obtained results showed that HepG2 cells exposed to 4, 10 and 25 μg/mL GO NPs displayed similar levels of LDH release compared to the untreated cells during the whole cultivation period (Figure 4A).

A statistically significant increase in LDH release (*p* < 0.001) was noticed after 24 h of exposure of HepG2 cells to haGO-NH_2_ NPs which however was not found to be concentration-dependent. Interestingly, we have found a decrease in LDH levels in GO-treated cells with concentration of 50 μg/mL. Analysis of LDH leakage revealed that only aminated GO NPs affect cell membrane integrity, which possibly induce cytotoxicity in HepG2 cells.

Cell membrane integrity after 24 h exposure to GO and haGO-NH_2_ NPs was qualified by FDA staining. FDA is a non-polar and non-fluorescent molecule, which enters the cell. Inside, it is hydrolyzed by intracellular cell esterases, and fluorescein is produced. This polar compound cannot leave the viable cell because it is unable to pass through the intact cell membrane, and accumulates in the cytoplasm of the cell and exhibits green fluorescence. Damaged cells, however, cannot retain the fluorescein, and they fluoresce very poor or are unstained. Fluorescent images on Figure 4B clearly show that the number of viable cells is reduced in haGO-NH_2_ treated samples suggesting the haGO-NH_2_ compromised in a greater degree the cell membrane than GO, which results in cell detachment and death.

### 2.3. Elevated Oxidative Stress in HepG2 Cells Detected after Incubation with haGO-NH_2_

Another possible mechanism for induction of cytotoxicity in HepG2 cells after incubation with the tested nanoparticles could be the elevated production of reactive oxygen species (ROS) leading to increased oxidative stress. ROS are by-products of biochemical reactions like mitochondrial respiration and cytochrome P450 enzymatic metabolism which have the potential to cause oxidative stress and damage in bio-molecules like lipids, proteins and DNA when ROS levels increase. Nanoparticles are known to initiate oxidative stress directly or indirectly through various mechanisms, thus exerting negative biological effects [33]. To verify the effects of tested GO NPs on oxidative stress, HepG2 cells were treated with both types of GO NPs for 24 h and ROS levels were then measured using enzymatic cleavage of DCFH-DA. As shown in Figure 5, HepG2 cells treated with both types of GOs NPs demonstrated a dose-dependent increase in ROS production. However, only the highest concentration of pristine GO (50 μg/mL) induced higher ROS production than the control cells. Inversely, all tested concentrations of haGO-NH_2_ induced production of much higher ROS levels than those measured in non-treated cells and in GO treated cells. This indicated that haGO-NH_2_ may potentially cause oxidative stress, which could impair normal physiological redox-regulated functions and thus induce cell death as detected in the previous experiments measuring cytotoxicity.

### 2.4. Both Types of Graphene Oxide Nanoparticles (GO and haGO-NH_2_) Trigger Mitochondrial Dysfunction in HepG2 Cells

One major source of increased cellular ROS levels is dysfunctional mitochondria. The mitochondrial oxygen consumption rate (OCR), which is a key metric of aerobic mitochondrial function, and the extracellular acidification rate (ECAR), which approximates glycolytic activity, were analyzed simultaneously using a standard mitochondrial stress test paradigm on a Seahorse analyser. The Seahorse analyzer permits to measure oxidative phosphorylation in a more physiologically relevant context. We estimated OCR and ECAR in HepG2 cells, treated with pristine and aminated graphene oxide NPs, for 24 h. Initially, we measured the basal respiration, and then, respiration after sequential injection of oligomycin, FCCP and antimycin. Oligomycin blocks ATP synthase activity and enables mitochondrial ATP production to be evaluated. FCCP is a powerful OxPhos uncoupler, which uncouples ATP synthesis from the ETC to dissipate the mitochondrial membrane potential and assess maximal mitochondrial activity independently of ATP production. Antimycin blocks residual mitochondrial activity to account for non-mitochondrial oxygen consumption. Measuring the change in concentrations of oxygen (O_2_) and free proton (H^+^), in the extracellular media over a prescribed time frame, provides data about the oxygen consumption rate (OCR, pmol/min) and extracellular acidification rate (ECAR pmol/min). As shown in Figure 6A, the mitochondrial respiration of HepG2 cells was compromised by both types of GO NPs. The toxic effects of GO and haGO-NH_2_ on HepG2 cells resulted in a decreased basal OCR, ATP-linked respiration, proton leakage and maximal respiration in comparison to the non-treated controls. Both types of GO NPs demonstrated a dose-dependent effect on basal OCR, ATP-linked respiration and proton leakage. However, our results show that they influenced the components of OCR differently and the effect of haGO-NH_2_ on mitochondrial functions was stronger compared to pristine GO NPs. GO and haGO-NH_2_ NPs decreased mitochondrial respiration of HepG2 cells even at the basal state (Figure 6B). Treatment of HepG2 cells with 4 and 10 μg/mL of GO and haGO-NH_2_ had small effect on basal respiration, while the highest concentrations of 25 and 50 μg/mL reduced basal respiration significantly compared to the control samples. In general, the basal OCR is composed of respiration linked to ATP production and proton leakage. ATP-linked respiration (Basal OCR - Oligomycin response) was significantly lower (*p*< 0.05) in both, 25 and 50 μg/mL GO and haGO-NH_2_ treated groups: 30 and 17.85 pmol/min as well as 29 and 25 pmol/min versus both controls 57.75 and 49 pmol/min. Proton leak (Antimycin A and Rotenone response - Oligomycin response) was reduced in a greater degree in 50 μg/mL GO and haGO-NH_2_ treated samples: 15, and 20 pmol/mL, respectively versus 28 and 31 pmol/mL for the controls, in a lower degree in 4, 10 and 25 μg/mL haGO-NH_2_-treated samples. However, in 4 μg/mL GO-treated cells, it was similar to the normal level (31 versus 28 pmol/ min). Changes in the proton leak pathway affect respiration rate - the increased proton leak uncouples oxidation and phosphorylation, i.e., decreases coupling efficiency. We found that the coupling efficiency decreases in both cases - after GO (from 68 to 54%) and after haGO-NH_2_ treatment (from 73 to 61%) (Figure 6B). This uncoupling very possibly reduced ATP production.

The maximal respiratory capacity was estimated by an FCCP-stimulated respiration and the observed decrease was a strong indicator of potential mitochondrial dysfunction. The experiments, conducted with both GO types, indicated that GO and haGO-NH_2_ caused perturbations in mitochondrial respiration. In GO-treated cells, the maximum respiration decreased in a dose-dependent manner, while in HepG2 cells-treated with 4, 25 and 50 μg/mL haGO-NH_2_ had a similar effect. However, in cells treated with 10 μg/mL haGO-NH_2_, an even weaker stimulating effect was observed. The cell spare respiratory capacity (SRC), however, was not impaired after 24 h-treatment with GO and haGO-NH_2_ NPs. The mitochondrial SRC is regarded as an important aspect of the mitochondrial function and is calculated by the difference between maximal and basal cellular OCR. When cells are subjected to stress, energy demands increase, with more ATP required to maintain cellular functions. A cell with a larger spare respiratory capacity can produce more ATP and overcome stress more effectively, which indicates that this could estimate the cells’ ability to cope with large increases in ATP turnover [34]. Consequently, GOs exposure, which negatively affects mitochondrial function, possibly exerts negative effects on the ability of cells to cope with other stress. Finally, the addition of a potent respiratory chain inhibitor, such as antimycin A, allows the estimation of non-mitochondrial OCR. In GO-treated samples, non-mitochondrial oxygen consumption decreased with increasing of GO concentrations, but without statistical significance. While, the effect in haGO-NH_2_ treated cells was exactly the opposite. There was a slight increase in this parameter with the increase of NPs concentrations, statistically significant only in cells treated with the highest concentrations of 50 μg/mL haGO-NH_2_ (Figure 6B).

Additionally, we have used the extracellular acidification rate (ECAR) as a proxy to evaluate glycolytic activity. We have calculated the rate between OCR and ECAR and have found that cells treated with both types of NPs had a lower basal OCR to ECAR ratio than the control cells (Figure 6C), suggesting that they rely on glycolysis rather than on OxPhos for ATP production.

### 2.5. Pristine and Aminated GO (GO and haGO-NH_2_) Prove Non-Genotoxic for HepG2 Cells

In order to dissect the mechanism of cytotoxicity of the tested pristine and aminated graphene oxide NPs on HepG2 cells we have performed Comet Assay, also called single-cell gel electrophoresis (SCGE). The Comet Assay sensitively detects damages in DNA [35,36,37]. HepG2 cells were treated with increasing concentrations of GO and haGO-NH_2_ (4, 10, 25 and 50 µg/mL) for 24 h at optimal conditions and were subjected to Comet Assay. HepG2 cells treated with 5 mM H_2_O_2_ for 30 min at 37 °C were used as a positive control for genotoxicity. Genotoxicity was further quantified by the software program CometScore and results are shown on Figure 7. “Comet Length” is a parameter in SCGE data analysis that gives representative and precise estimation of the level of genotoxicity of the tested substances. HepG2 cells treated with pristine GO showed very faint almost insignificant presence of DNA damage when incubated for 24 h. On the contrary, the haGO-NH_2_ displayed slightly higher genotoxicity effect on the cells but surprisingly this was detected at the lowest used concentration of 4 and 10 µg/mL. The given trendline (red dotted line on Figure 7) represents the moving average values for Comet length measured for all probes, including the positive control for genotoxicity - HepG2 cells treated with 5mM H_2_O_2_. It displays the presence of very faint genotoxic potential of GO and haGO-NH_2_ on HepG2 cells, pointing out the mechanisms by which graphene oxide GO NPs exert their biological activities are not centered on the stability and maintenance of genome integrity.

## 3. Discussion

In this work, we studied the efficacy of newly synthesized aminated (haGO-NH_2_) and pristine graphene oxide nanoparticles as potential new anticancer agents for treating hepatocellular carcinoma cells. We developed a new protocol for amination of GO with hydroxylamine, which proved to be simple, cost and time-effective, and moreover, quite successful, as proven by the physico-chemical characterization of the new materials. In our previous research, we investigated commercially available aminated by ammonia graphene oxide nanoparticles (GO-NH_2_) as anticancer agents for colorectal cancer cells. We found that GO-NH_2_ NPs trigger stronger cytotoxic and genotoxic effect than pristine GO by induction of ROS, DNA damage and apoptosis in Colon 26 cells [20]. In order to continue with the analysis of the biological effects of aminated GO nanoparticles on other cell types, and to investigate, in detail, the mechanism of nanoparticle action, we developed a new protocol for the synthesis of GO-NH_2_ NPs using hydroxylamine as a new reducing agent. The physicochemical characterization of the newly-synthesized hydroxylamine modified haGO-NH_2_ NPs demonstrated that amination by hydroxylamine has a similar modification effect as ammonia on the size and morphology of GO NPs. Both methods of amination led to increased wrinkling of the nanosheets and decreased size of the particles. We have summarized data from physico-chemical characterization together with the biological activity of aminated GO NPs, by both methods in the table below (Table 1). As demonstrated in the table, both aminated types of NPs had very similar size of 560nm for GO-NH_2_ versus 594 nm for haGO-NH_2_. However, a difference in respect to the measured zeta potential between both types of aminated GO NPs was observed. The aminated by hydroxylamine haGO-NH_2_ had a negative charge (−12.28 eV), while the commercially available GO-NH_2_ had a positive charge of at 38.5 eV. We suggest that this difference could be referred to as the difference in the total amount of nitrogen in both samples, i.e. 1.86% in hydroxylamine-aminated haGO-NH_2_ versus 3.47% in ammonia-modified GO-NH_2_ NPs and could result in different interactions with cells. When compared the biological activity of haGO-NH_2_ and ammonia-modified GO-NH_2_ we have established a reduced cell adhesion ability after exposure to 50 μg/mL NPs, similar in both studied cell types - 21% in HepG2 cells and 22.5% in Colon 26 cells. The effect of haGO-NH_2_nanoparticles on cell adhesion was stronger than that of ammonia-modified GO-NH_2_ NPs when compared to GO, because in haGO-NH_2_ –treated HepG2 cells adhesion was 21% versus 63% in GO-treated cells while in ammonia-treated Colon 26 cells adhesion was 22.5% versus 42.6% in GO-treated Colon 26 cells. A greater reduction in the inhibitory concentration, IC_50_, of haGO-NH_2_ was also found, suggesting that amination by hydroxylamine is more effective than amination by ammonia. It should be kept in mind however the different type of the studied cells as well as the different tests used for calculating the % of cell adhesion on which the calculation of IC50 is based. The case based on Colon 26 cells is based on counting the number of attached cells by an automated cell counter (Countess, Invitrogen), while for HepG2 cells, the assay of WST-1.

Biological activities of the newly synthesized aminated GO NPs were first assessed by evaluating HepG2 morphology and viability by means of Neutral Red staining and WST-1 assay. Generally, cell morphology together with cell viability are essential signs of the physiological status of the cells as both are important indicators for cytotoxicity. In our study, we found that the rate of survival of HepG2 cells, treated with pristine and aminated GO NPs, decreased with the increase in the concentrations of both types of nanomaterials, and dropped down to 80% after treatment with the highest concentration used, i.e.,50 μg/mL, especially in the case with haGO-NH_2_. This suggests high cellular toxicity, especially of the aminated GO NPs. The morphological observations under the light microscope did not disclose any significant differences in cell morphology. This is in contrast to other studies, including ours [20,38], that report pronounced morphological alterations, and the appearance of apoptotic-like morphology when GO and GO-NH_2_ were used to treat macrophages and colorectal carcinoma cells. The difference in the results could be related to the different types of cells used in the experiments, and further to the different protocols for modification of the graphene oxide nanomaterials. Further, we measured LDH leakage in the cell culture medium and found an increased amount of LDH after exposure of HepG2 cells, only to haGO-NH_2_ NPs, thus, suggesting damage in plasma membrane structural integrity. LDH is a relatively stable intracellular enzyme which can leak out only when the cell membrane is broken. Our results suggest that haGO-NH_2_ penetrates through the plasma membrane and probably disrupts the phospholipid bilayer unlike GO NPs. The last could be due to the fact that pristine GO NPs are larger in size, which possibly hinders penetration through the cell membrane. The results are in agreement with Chang et al. [11] who have measured a lower LDH leakage in A459 cells, treated with GO nanoparticles, with concentrations 50 μg/mL and above, compared to the control. A study by Sasidharan et al. compared carboxyl-functionalized graphene with pristine graphene, and found that no LDH leakage was observed at concentrations as high as 300 μg/mL in Vero cells [39]. Moreover, Zhang et al. [40] observed that graphene aggregates were attached to the surface of rat PC12 cells and caused an increase in LDH leakage only at the highest exposure concentration (100 μg/mL). On the contrary, Liao et al. demonstrated that both, pristine graphene and GO sheets were able to disrupt the plasma membrane of erythrocytes [41]. The discrepancy in the results might be explained with the different types of the cells used, different types of GO, as well as the different experimental design. It should be considered also the different mechanism of toxicity induced by NPs, and that LDH release is a marker of necrotic cell death [42,43]. Therefore, if the GO NPs induced apoptotic cell death without cell membrane damage, then LDH is not released in cell culture medium, and thus, cannot be measured.

Little is known about the mechanisms by which aminated graphene oxide nanoparticles induce toxicity. Some authors hypothesize that induction of cellular oxidative stress is considered as one of the mechanisms underlying nanomaterial toxicity in general. Commonly, oxidative stress results from the imbalance between oxidative and antioxidative defense systems of cells and tissues, and a result of overproduction of oxidative-free radicals and ROS. An outcome of excessive levels of ROS is the modification of the structure and function of cellular proteins and lipids, leading to cellular dysfunction, including impaired energy metabolism, altered cell signaling and cell cycle control, impaired cell transport mechanisms and overall dysfunctional biological activity, immune activation and inflammation [44,45,46]. Therefore, we measured the generation of ROS by studying the mechanisms of detected hepatotoxicity-induced by GO NPs. In our study, GO and haGO-NH_2_ nanoparticles were observed to induce generation of intracellular ROS in a concentration-dependent manner. In addition, GO and haGO-NH_2_ ROS production seemed to follow different trends. For GO, maximum ROS levels were reached after exposure to 50 μg/mL after 24 h of exposure of the cells to the nanoparticles. In cells treated with lower GO concentrations (between 4 and 25 μg/mL), intracellular ROS levels were kept even lower than the control and eventually reached levels comparable to those measured for the control (especially at a concentration of 25 μg/mL). On the contrary, 24 h of exposure to the increasing concentrations of haGO-NH_2_ (4–25μg/mL) resulted in significant increment in ROS levels while at the highest concentration of 50 μg/mL, the detected ROS levels declined. Regarding the oxidant-generating potential of haGO-NH_2_, the obtained results are consistent with our previous observations for colorectal cancer cells where a bell-shaped curve of ROS production was shown for both types of GO (pristine and aminated) [20]. A possible explanation could be that GO particles have high adsorption potential, which may cause quenching of the signal by depleting the fluorophore, and thus, producing false signals. This very possibly could result in reduced ROS production at the highest concentrations of GO and GO-NH_2_. Our results in respect of ROS production, induced by pristine GO NPs in HepG2 cells, are in accordance with those reported by other authors like Yuan et al. [47] who did not detect any significant increase in the intracellular ROS levels in HepG2 cells, exposed to 1 μg/mL of single-layered GO. Based on the obtained results here, and those in our previous study on ROS production in colon cancer cells, we conclude that the effect of GO on ROS formation is rather cell-specific. These conclusions are also supported by literature data on the ability of GO to induce the generation of intracellular ROS in other cell lines. A549 cells exposed to 10 μg/mL GO for 24 h, in comparable ROS levels to those determined in this study [48]. In human skin fibroblasts, however, no significant increase with respect to the control could be detected after 24 h of exposure to concentrations as high as 25 μg/mL [49]. The discrepancy between the results obtained in this study and those stated above (including ours) might be due to differences in the lateral size of the nanoparticles tested (>1 μm), the suspension protocol (serum-free medium), the assay protocol (loading of the cells with the dye DCFH-DA was carried out prior to treatment), not only due to the sensitivity of the used cell lines. To the best of our knowledge, no data on the oxidant-generating ability of haGO-NH_2_ have been reported in the scientific literature. The fact that GO and haGO-NH_2_-induced ROS generation displayed different kinetics suggests that the underlying ROS generating mechanisms are distinct. The exact mechanism(s) by which a nanomaterial exerts oxidative stress is relatively difficult to be identified, and still remains to be elucidated for most nanomaterials, including graphene and its derivates. An integrative consideration of the results obtained by different assays can assist in obtaining the first indication on the possible mechanisms involved. In general, there are two possible mechanisms for ROS induction: Direct and indirect. Direct ROS generation typically involves processes that are independent of the presence of biological systems (namely acellular ROS generation), i.e., are solely a function of the nanomaterial’s physico-chemical properties. Indirect ROS generation, on the contrary, typically involves cellular (i.e., biochemical) processes that were triggered by the nanomaterial beforehand [50].

Mitochondria perform essential functions in generating most of the cellular energy through the oxidative phosphorylation system and important metabolic intermediates in various pathways, such as amino acids, fatty acids and carbohydrates. It is known that mitochondria play compelling roles in carcinogenesis via altered energy metabolism, resistance to apoptosis, increased production of ROS and mtDNA (mitochondrial genome) changes [51,52,53,54,55]. ROS produced by nanoparticle-induced damage of the respiratory chain, likely by disturbing mitochondrial membrane permeability. When we investigated the nanomaterials’ effect on cellular respiration in mitochondria, we found that the exposure to GO and haGO-NH_2_ nanoparticles resulted in the suppression of most of the phases of cellular respiration. Mitochondrial dysfunction is known to be associated with oxidative damage of mitochondrial macromolecules including mtDNA, lipids and proteins. [54]. Damage of mitochondrial functions can be provoked directly, i.e., by physical interaction of the nanomaterials with the mitochondrial membrane, or indirectly, e.g., by inducing damages through interactions with cell membrane protein receptors [56,57,58,59]. It is known that DNA is a critical target of ROS. A higher production of intracellular ROS may lead to the oxidative damage of DNA, including base and sugar lesions, DNA–protein crosslinks, single- and double-strand breaks, and the formation of alkali-labile sites [29]. In our previous studies, we have shown that ammonia-modified GO NPs have the potential to induce DNA damage and apoptosis in Colon 26 cancer cells. Hence, we have analysed here the newly synthesized haGO-NH_2_ NPs and their ability to destroy DNA in HepG2 cells. We found that hydroxylamine aminated GO induced only slight DNA damage, although caused ROS generation and mitochondrial dysfunction. This suggests that the mechanisms by which haGO-NH_2_NPs exert their biological activities are not centered on the stability and maintenance of the genome integrity, but rather in mitochondrial metabolism and oxidative damage. The latter is in contrast to the observed DNA damage in ammonia-modified GO in colon cancer cells.

## 4. Materials and Methods

### 4.1. Amination of Graphene Oxide Particles

GO was purchased from Graphenea (C1576, San Sebastian, Spain) as a water suspension with a concentration of 4 mg/mL. Amination was achieved by the addition of 2 mL hydroxylamine (50% solution, Merck, Darmstadt, Germany) to 20 mL GO solution and was kept under continuous stirring on magnetic stir for 5 h at 80 °C. Unreacted materials were removed by three times washing with deionized water by centrifugation at 3000 rpm for 30 min. The supernatant was removed and deionized water was added to the sediment up to 20 mL. The optical density of the resulting product was measured at 270 nm wavelength in order to get the desired concentrations of aminated GO NPs.

Immediately before cell experiments, particle stock suspensions were sonicated in an ultrasonic water bath (50 Hz, UM-2, Unitra-Unima, Olsztyn, Poland) for 1 h. The desired final concentrations of NPs (4, 10, 25, and 50 μg/mL, respectively) were achieved by adding certain volume of the nanoparticles from the stock solutions directly into the culture medium.

TEM (JEM-2100, JEOL, Tokyo, Japan) images were acquired at 200 kV using Holey-carbon film on 300 mesh nickel grids. Prior to TEM imaging GO and GO-NH_2_ suspensions were sonicated for 60 min.

Dynamic light scattering (DLS) for characterization of size and zeta potential of the nanoparticles in solution, was performed on a Zetatrac instrument (S3500; Microtrac, Largo, FL, USA). Samples were examined after dilution of nanoparticle to a stock solution of 100 mg/mL suspensions in DI water, and sonicated for 1 h, then 1 mL was transferred to a Zetatrac instrument for DLS measurement.

### 4.2. Cell Culture

HepG2 cells were grown in MEM culture medium supplemented with 10% fetal bovine serum (FBS, Sigma-Aldrich, Germany) and an antibiotic–antimycotic solution (Sigma-Aldrich, Germany). The cells were grown in a humidified environment with 5% CO_2_ and 95% atmosphere at 37 °C. For in vitro experiments, the pre-confluent cells were detached using a mixture of 0.05% trypsin and 0.02% EDTA (Sigma-Aldrich, Germany) and were seeded at a density of 2 × 10^4^ cells/well in a 24-well plate or 1 × 10^5^ cells/well in 6-well plates depending on the protocol. Cells were cultivated for 24 h before exposure to increasing concentrations of GO and GO-NH_2_ nanoparticles. After adding the nanoparticles, the cells were incubated for another 24, 48 or 72 h and after that were processed according to the protocol. Control cells were processed in the same way as tested samples, but in the absence of nanoparticles.

### 4.3. Phase-Contrast Light and Fluorescent Microscopy

Phase-contrast light and fluorescent microscopy observations were done in order to evaluate alterations in cell morphology after 24 h of exposure to both types of GO NPs and the integrity of the cell membrane, respectively. The phase-contrast micrographs were taken at magnifications of 25× with a Leitz microscope equipped with a digital camera after staining of cells with neutral red (Sigma-Aldrich, Germany), while fluorescent micrographs were taken at magnification 10×, with an inverted microscope Axiovert 25 (Carl Zeiss, Germany), equipped with a digital camera after staining of cells with 0.001% fluorescein diacetate (FDA), as previously described [20].

### 4.4. WST-1 Assay

WST-1 (Sigma-Aldrich Co., Germany) was used to evaluate cell viability after 24 h exposure to GO NPs. It is a sensitive colorimetric assay using a water-soluble tetrazolium salt WST-1 (2-(2-methoxy-4-nitrophenyl)-3-(4-nitrophenyl)-5-(2,4-disulfophenyl)-2H-tetrazolium, monosodium salt) to quantify the number of live cells by producing an orange formazan dye upon bio-reduction in the presence of an electron carrier. Briefly, the WST-1 solution was added directly to the cells in ratio 1:10. After 4 h of incubation at 37 °C, at dark, the amount of the formazan produced was measured by absorbance at 450 nm. Cell viability is demonstrated as a graph with the measured optical density (OD) values, where values represent the MEAN ± STDV of three repetitive experiments.

IC_50_ values were calculated by means of GraphPad Prism 7 (GraphPad Software, San Diego, CA, USA) based on the data obtained from WST-1 assay – for HepG2 cells and from the number of attached cell, calculated by an automated cell counter (Countess, Invitrogen, USA) for Colon26 cells

### 4.5. LDH Assay

Membrane integrity was assessed by measuring extracellular lactate dehydrogenase (LDH), using a commercially available kit (LDH cytotoxicity detection kit, Roche Diagnostic, IN, USA). Cytosolic LDH is released from the cells into the culture medium if the integrity of the cell membrane deteriorates in case of irreversible cell death. Briefly, HepG2 cells were seeded in 24-well plates at a density of 2 × 10^4^ cells/mL culture medium. After 24 h of seeding, the culture medium was replaced with fresh one and the tested NPs with increasing concentrations were added to the wells. The plates were then incubated for 24 h at 37 °C under 100% humidity and 5% CO_2_. Cell-free culture media was collected. LDH activity was measured at 490 nm by UV-Visible absorbance microplate reader. Background and negative controls were obtained by LDH measurement of assay medium, and untreated cell medium, respectively. Total cellular LDH activity (positive control) was measured in cell lysates obtained by treatment with TritonX-100 solution. The OD_490nm_ of the cells permeabilized with Triton X-100, was accepted as 100% of LDH release. The percentage of LDH release in the cells treated with nanoparticles was calculated as a percentage and presented in a graph. Three repetitions of the experiment have been done and values are MEAN±STDV.

### 4.6. DCFA-DA Analysis

The production of intracellular reactive oxygen species (ROS) was measured using 2,7-dichlorofluorescin diacetate (DCFH-DA, Sigma-Aldrich, Germany) as described before [20,49]. The DCFH-DA is a non-fluorescent compound, which passively enters the cell and reacts with ROS to form the highly fluorescent compound dichlorofluorescin (DCF). In brief, HepG2 cells (3×10^4^) were seeded in 24-well plates and allowed for adherence. Following respective exposure, the cells were washed twice with PBS and incubated for 30 min in dark in FBS-free culture medium, containing DCFH-DA (20 μM). Then, the DCFH-DA containing medium was removed, the control (untreated) and the treated cells were rinsed twice with PBS, and the fluorescence intensity of DCF was detected on a spectrofluorometer upon excitation at 485 nm and emission at 520 nm. The results are presented as graph, where by the bars represent the MEAN values ±STDV of three experiment repetitions.

### 4.7. Single-Cell Gel Electrophoresis (SCGE)

SCGE was performed as previously described [29]. Briefly, 1 × 10^3^ cells were mixed with 0.7% (f.c.) of low-gelling agarose (Sigma-Aldrich, Germany) and were layered as microgels on microscopic slides. The slides were then lysed in 146 mM NaCl, 30 mM EDTA, pH 7, 10 mM Tris-HCl, pH-7 and 0.1% *N*-lauroylsarcosine (NLS, Sigma-Aldrich, Germany) at 10 °C for 20 min and were electrophoresed for 20 min at 0.46 V/cm. The results were visualized under a fluorescent microscope after staining of gels with SYBR green (Molecular probes, Invitrogen). The results were quantified by Comet Assay specialized software CometScore. HepG2 cells treated with 5 mM H_2_O_2_(Sigma-Aldrich, Germany) for 30 min at 37 °C were used as a positive control for genotoxicity. After treatment the cells were washed in 1xPBS buffer (2.68 mM KCl, 1.47 mM KH_2_PO_4_, 1.37 mM NaCl, 8 mM Na_2_HPO_4_), pH 7 and subjected to Comet Assay. Comet Assay data analysis was done by the software CometScore and results are presented as a graph on which the MEAN values of the parameter Comet length ±STDV are given as bars.

### 4.8. Mitochondrial Stress Analysis

The mitochondrial oxygen consumption rate (OCR) and the extracellular acidification rate (ECAR) were analyzed simultaneously using a standard mitochondrial stress test paradigm on the Seahorse Bioscience XFp analyzer (Agilent Technologies, CA, USA). The cells were assayed for OCR and ECAR measurements following the manufacturer’s instructions. Briefly, before analysis, cells were washed with unbuffered assay media (Seahorse XF DMEM, pH-7.4) without phenol red supplemented with glucose (10 mM), sodium pyruvate (1mM) and glutamine (2 mM), and incubated for 1 h in a CO_2_-free incubator at 37 °C. After the initial measurement of basal OCR and ECAR, the inhibitors of mitochondrial activity were injected sequentially in the ports on the cartridges. First, the inhibitor of ATP synthase oligomycin (1μM) was added to determine both parts of basal respiration - one, that is used to drive the ATP production and the other – independent proton leakage across the inner mitochondrial membrane. Next, the uncoupler of mitochondrial oxidative phosphorylation carbonyl cyanide-4-(trifuoromethoxy) phenylhydrazone (FCCP, 0.125μM) was added. It induces a collapse of the inner membrane gradient, driving the mitochondria to respire at their maximal rate. Finally, the complex III inhibitor, antimycin A (1 μM) together with complex I inhibitor rotenone (1 μM), an inhibitor of mitochondrial NADH dehydrogenase, were added to determine non-mitochondrial respiration. Basal respiration or acidification was calculated using the mean of the three OCR or ECAR measurements, before the first injection. ATP-production, proton leak, as well as maximal respiration were calculated as the mean of three OCR measurement cycles after oligomycin, or FCCP injection, respectively. Maximal acidification was calculated as the mean of three ECAR measurement cycles after oligomycin injection. The OCR data were corrected for non-mitochondrial oxygen consumption under rotenone and antimycin A [59]. Two repetitions of the Seahorse experiments were performed for data quantitation.

### 4.9. Statistical Analysis

Data in this article were statistically analyzed by the Microsoft Excel software in which bars represent the MEAN values of the calculated parameters ± STDV. Additionally, Student’s *t*-test, where the probability levels of 0.05 were considered as statistically significant. Additionally, linear regression models were calculated for WST-1experiments for evaluation of cytotoxicity of the tested NPs on HepG2 cells.

## 5. Conclusions

In the present study, we developed a simple, effective, cheap and time-saving protocol for amination of GO by hydroxylamine. Hydroxylamine-aminated GONPs (haGO-NH_2_) were characterized by XPS, TEM, and Zetasizer. The results demonstrated that amination of GO by hydroxylamine decreased the size and zeta potential, but increased the wrinkles of the GO sheets. The cytotoxic responses to the newly synthesized aminated GO and the underlying mechanisms were investigated in hepatocellular carcinoma HepG2 cells. We observed that exposure to haGO-NH_2_ significant induced cytotoxicity (reduced cell viability in a dose-dependent manner and cell membrane damage) and oxidative stress (increased ROS production and mitochondrial dysfunction) in HepG2 cells. No significant alterations in cell morphology, nor substantial DNA damage, were detected, compared to pristine GO and the control group. In conclusion, ROS production and extracellular acidification rate in mitochondria of haGO-NH_2_ treated cells could be a potential mechanism for cytotoxicity of hydroxylamine modified GO nanomaterials. These findings provide toxicological and mechanistic information that could enrich knowledge on molecular mechanisms exerted by pristine and modified GO nanomaterials in different biological systems. Their potential use as anticancer drugs and as vector delivery systems in cancer cells remains to be further elucidated.

## Figures and Tables

**Figure 1 ijms-21-02427-f001:**
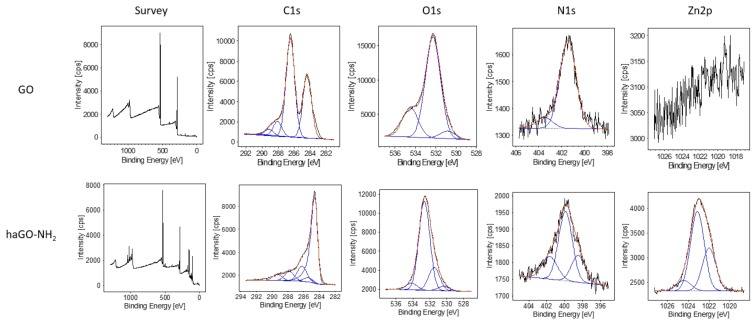
X-ray Photoemission Spectroscopy.The elemental composition of the nanoparticles was analyzed by X-ray photoemission spectroscopy (XPS) on Axis DLD Ultra instrument (Kratos–Manchester, UK).

**Figure 2 ijms-21-02427-f002:**
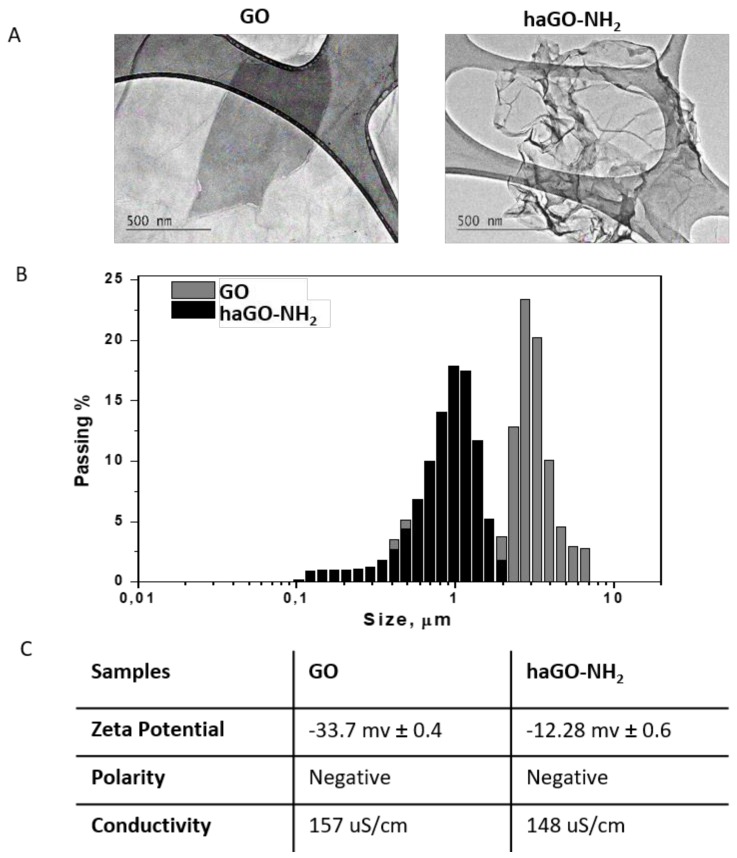
Biophysical characterization of GO and haGO-NH_2_ nanoparticles. (**A**) TEM micrographs of GO and haGO-NH_2_. Images were acquired at 200 kV using Holey-carbon film on 300 mesh nickel grids. (**B**,**C**) Characterization of size and zeta potential of the nanoparticles in solution, were performed on a Zetatrac instrument (S3500; Microtrac, Largo, FL).

**Figure 3 ijms-21-02427-f003:**
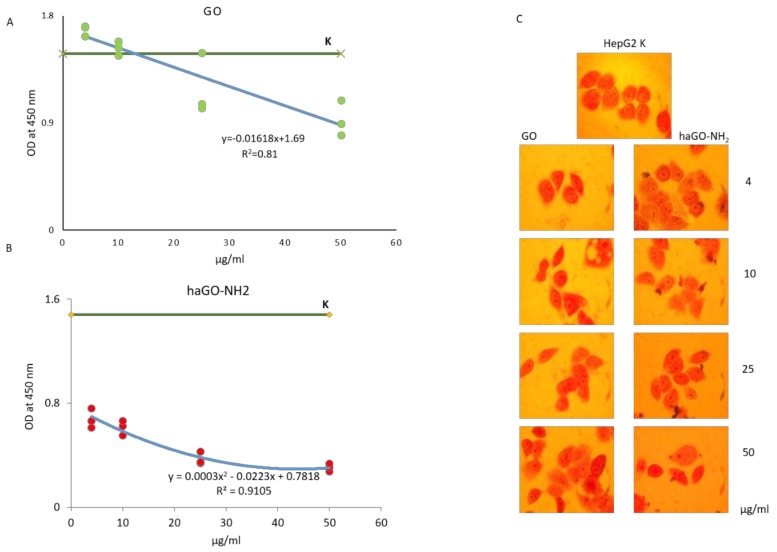
Cell viability and morphology of HepG2 cells treated for 24 h with GO and haGO-NH_2_. (**A**,**B**) WST-8 (Sigma-Aldrich Co.) was used to evaluate cell viability - linear regression models for both types of GO NPs cytotoxic effect on HepG2 cells. (**C**) A panel of micrographs with neutral red stained HepG2 cells taken under phase contrast microscopy after 24 h of incubation of the cells with different concentrations of GO and haGO-NH_2_. Magnification 25×.

**Figure 4 ijms-21-02427-f004:**
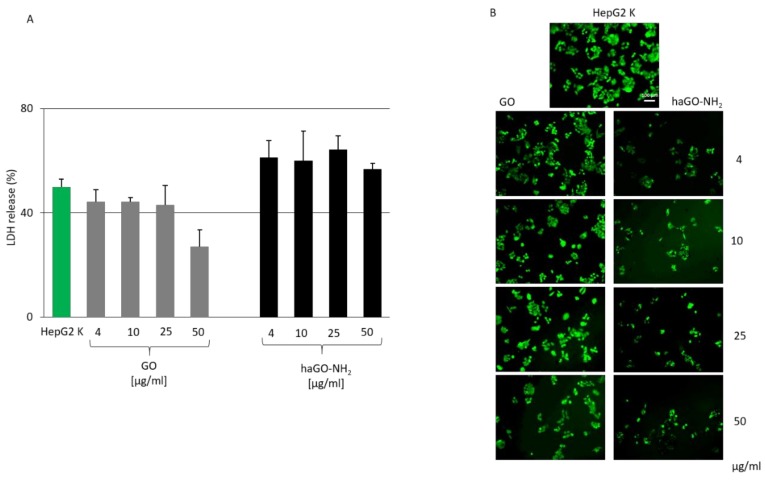
Membrane integrity of HepG2 cells treated for 24 h with pristine and aminated GO. (**A**) Percentage of LDH release fromHepG2 cells after 24 h of incubation in the presenceof different concentrations of GO and haGO-NH_2_ nanoparticles. Cells treated with Triton X-100 were used as a positive control. The LDH release values were quantified as a percentage of the LDH release in Triton X-100 treated cells which was taken as 100%. Values are MEAN ± STDV from three repetitive experiments. (**B**) A panel of FDA-stained HepG2 cells taken under a fluorescent microscope after the cells were treated with increasing concentrations of GO and haGO-NH_2_ nanoparticles for 24 h. Magnification 10×; bar 100 μm.

**Figure 5 ijms-21-02427-f005:**
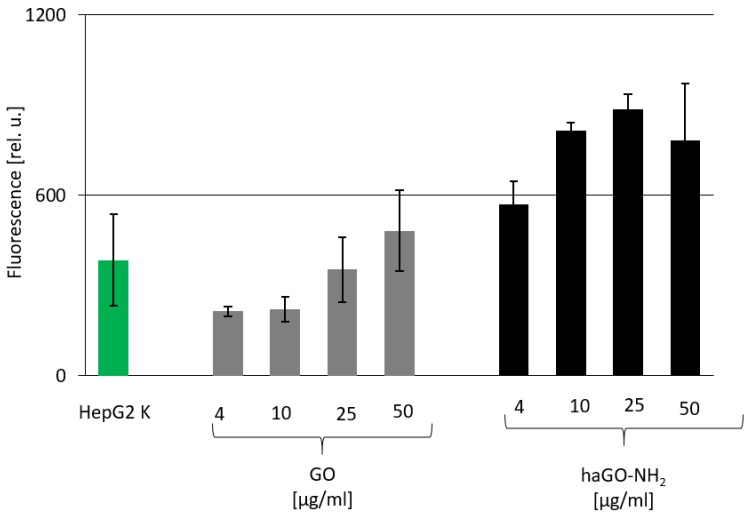
ROS production in HepG2 cells after treatment with GO nanoparticles. The production of intracellular ROS was measured using 2,7-dichlorofluorescin diacetate. HepG2 cells were seeded in 24-well plates and allowed for adherence. The fluorescence intensity of DCF was detected on a spectrofluorometer upon excitation at 485 nm and emission at 520 nm.

**Figure 6 ijms-21-02427-f006:**
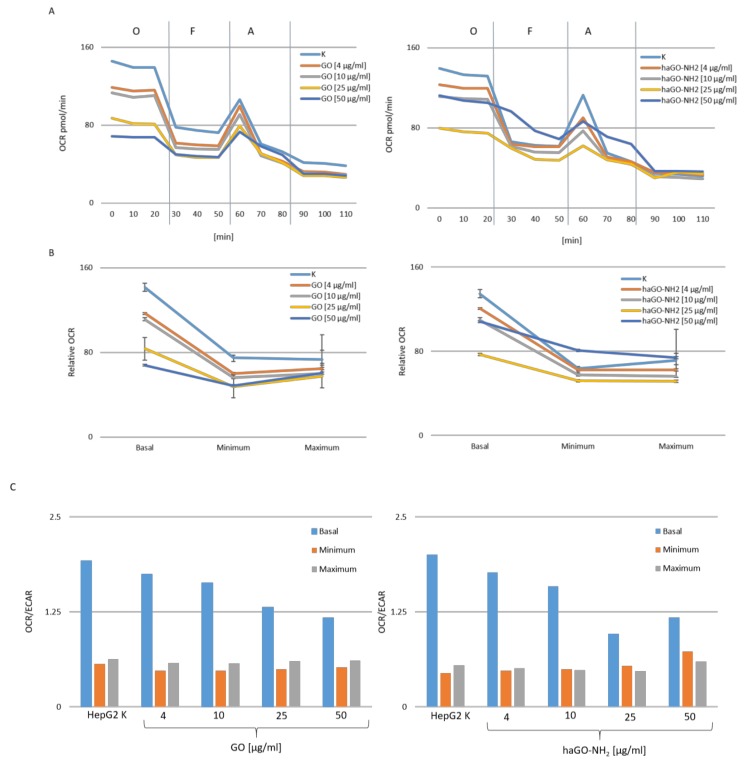
Metabolic studies of HepG2 cells treated with GO and haGO-NH_2_ NPs by Seahorse analyses. (**A**) Mitochondrial oxygen consumption rate (OCR) of HepG2 cells treated with pristine and aminated GO NPs for 24 h in real time under basal conditions and in response to mitochondrial inhibitors (O, oligomycin; F, FCCP; A, antimycin). (**B**) Mitochondrial parameters of cells treated for 24 h with GO and haGO-NH_2_. (**C**) Representation of the ratio between the basal OCR and ECAR where the OCR was measured at the same time as ECAR for HepG2 cells after treatment with both types of NPs.

**Figure 7 ijms-21-02427-f007:**
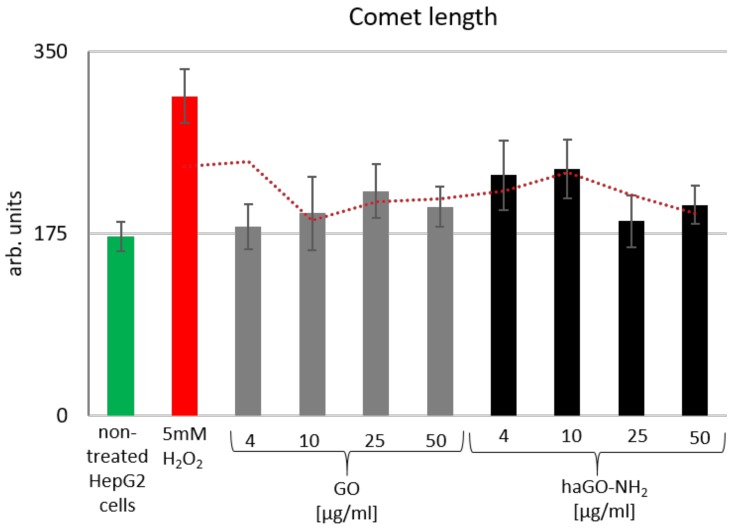
SCGE for testing the genotoxicity potential of pristine and aminated GO nanoparticles on HepG2 cells. Graphical representation of the parameter “Comet length” as quantified by the software CometScore. Data are represented as MEAN±STDV, where *n*=100. Additionally, the given trend represents the moving average values for Comet length measured for all probes including the positive control for genotoxicity - HepG2 cells treated with 5 mM H_2_O_2_.

**Table 1 ijms-21-02427-t001:** Comparison between different types of GO and aminated GO NPs studied in this study and in Krasteva et al., 2019 [20].

Sample	Mean Size	ZP (mV) ± SE	Polarity	N1sTOT.(%)	IC_50_HepG2 Cells	IC_50_Colon 26 Cells
GO(ref. [20])	250 ± 68 nm	−24.5 ± 0.4 mV	negative	0.99		1.71 ± 0.2 µg/mL
1.5 ± 0.7 μm
(this study)	515 ± 50 nm	−33.7 ± 0.4 mV			62.97 ± 10 µg/mL	
3.6 ± 0.5 μm
GO-NH_2_(ref. [20])	560 ± 300 nm	38.5 ± 2.8 mV	positive	3.47		1.26 ± 0.1 µg/mL
haGO-NH_2_hydroxylamine modified (this study)	594 ± 270 nm	−12.28 ± 0.6 mV	negative	1.86	3.4 ± 0.7µg/mL

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
