# Peer review of "Amination of Graphene Oxide Leads to Increased Cytotoxicity in Hepatocellular Carcinoma Cells"

_ijms, 2020, doi:10.3390/ijms21072427_

Round 1

Reviewer 1 Report

In the present paper Georgieva and colleagues described cytotoxic effects of NH2-GO nanoparticles in HepG2 cell line. They showed that the cytotoxic effects of NH2-GO NPs depend on cell membrane damage, mitochondrial dysfunction and increased ROS production. The following major points need to be addressed before publication.

As a biologist, it is difficult to follow why the authors use the same name GO-NH2 for amine-functionalized graphene oxide nanoparticles, they studied in their previous work (ref. 21) and the “novel-aminated graphene oxide” they described in the present work. They referred to the preparation of the new nanoparticles with graphene oxide and hydroxylamine but using the same name will not help to distinguish the two products that differ in size, thus, membrane penetration potential. It is mandatory to better specify and discuss the differences and probably change the name of the two different nanoparticles will help.

I believe that the most interesting comparison in term of biological activity should be performed between GO-NH2 NPs (ref. 21) and the newly synthetized NPs presented in the present paper. Authors try to discuss these differences in the discussion section, but it is quite misleading if it is possible to compare just physico-chemical characteristics and not their biological meaning on the same cellular model. Authors scarcely discussed this point (lines 323-327) without providing a convincing justification on the different activity observed in the two cell models.

Why the authors decide to switch from colon cancer to hepatocarcinoma cell model? The cellular mechanisms they analyzed are general, non-cell specific and for this reason they could have used the same cell model to compare the two different GO-NH2 NPs. Please comment.

It is not possible to appreciate differences in cell morphology according to the images published by the authors, due to poor magnification and the lack of a consistent analysis of morphological markers of cytotoxicity. It is clear that this microscopic analysis is not enough to see any difference. More, how the authors justify the different results recorded by WST-8 and LDH assay? They speculate about the linear regression models they adopted for the two nanoparticles GO and GO-NH2, but probably presenting cell viability results in percentage compared to untreated cells will be more self-explanatory for readers. Furthermore they could provide IC50 values.

Did Authors collect some data on apoptotic events recorded after GO and GO-NH2 treatment? They listed apoptosis in keywords section, but specific results are missing.

Authors discussed LDH literature data (328-342), without providing their conclusion, meaning how their data contribute to clarify contrasting evidence. Please clarify.

It would be interesting to know ROS kinetic, what happens at shorter time point? In the discussion section Authors discussed about different kinetic according to concentration, but the real kinetic would be the presentation of data at different time points (1, 3, 6 hours, together with 24).

Due to hypothesized experimental limits in detection of ROS level, it is possible to ascertain the role of ROS in the cytotoxic effects of GO-NH2 using an indirect method, e.g. pre-treatment with N-acetylcysteine will help to confirm the role of ROS in the cytotoxic effects of GO-NH2.

I believe that the conclusions on the genotoxic potential of GO-NH2 reported at the end of discussion section are scarcely discussed. Authors declared that data complies with published data on ammonia-modified GO obtained in colon cancer cells, where they assess genotoxicity of these NPs (ref.21). The different characteristics of the nanoparticles could justify a different behavior in term of genotoxicity? Both NPs induced ROS generation, so are differences imputable to the different cell model? Also in this case, the possibility to compare the different GO-NH2 NPs on the same model could had conferred additional value to presented results.

Authors discussed about “selectivity” reporting this term in the title and abstract section, but I believe this term is misleading. Selectivity would mean activity of NPs assessed in cancer cells versus healthy cells (i.e. non transformed hepatic cells). Here they showed that GO-NH2 NPs had highest cytotoxic activity compared to GO NPs. Please clarify.

Minor points:

It is not possible to completely read legend of Figure 2.

Figure 5: why error bars are not present in all columns?

Positive control for comet assay is missing.

Author Response

 Dear  Prof. Maurizio Battino,

On behalf of my co-authors and me I would like to submit our revised manuscript entitled: “Amination of graphene oxide leads to increased cytotoxicity in hepatocellular carcinoma cells” to the “International Journal of Molecular Science.

We have followed all reviewers’ comments and after major revision, and additional experiments and data evaluation we now believe that the manuscript is more advanced and far more comprehensible.

With this we express our confidence that all the efforts by us and the reviewers will be evaluated positively and our manuscript will be published in your journal.

Please, find below our detailed answers to all remarks, concerns and questions.

Best regards,

Assoc. Prof. Natalia Krasteva, PhD

Institute of Biophysics and Biomedical Engineering

Bulgarian Academy of Sciences

“Acad. G. Bonchev” str., bl. 21

1113 Sofia, Bulgaria

tel: +359 2 979 26 43

email: [email protected]

Reviewer 1:

Submission Date

05 February 2020

Date of this review

23 Feb 2020 23:56:21

Comment 1:

“In the present paper Georgieva and colleagues described cytotoxic effects of NH2-GO nanoparticles in HepG2 cell line. They showed that the cytotoxic effects of NH2-GO NPs depend on cell membrane damage, mitochondrial dysfunction and increased ROS production. The following major points need to be addressed before publication.

As a biologist, it is difficult to follow why the authors use the same name GO-NH2 for amine-functionalized graphene oxide nanoparticles, they studied in their previous work (ref. 21) and the “novel-aminated graphene oxide” they described in the present work. They referred to the preparation of the new nanoparticles with graphene oxide and hydroxylamine but using the same name will not help to distinguish the two products that differ in size, thus, membrane penetration potential. It is mandatory to better specify and discuss the differences and probably change the name of the two different nanoparticles will help.

Answer:  

The following comment: ”it is mandatory to better specify and discuss the differences and probably change the name of the two different nanoparticles will helpis really meaningful. We thank the reviewer for this. It made us decide to distinguish between the differently modified GO nanoparticles by, first, changing the name of hydroxylamine modified ones with the acronym haGO-NH2 and secondly, by building a comparative table (Table 1, line 386 in the text). The Table is included in the Discussion section and we find it very useful. It helps the reader to follow the logic of our discussion and to easily compare the properties of GOs modified by different protocols: those modified by hydroxylamine and reported in this study and those modified by ammonia already published in Krasteva et al., 2019 (pls. see ref. 21). The text in the Discussion section appears clearer and far more comprehensible, especially when we compare the properties of all tested graphene oxide NPs studied in this study and in Krasteva et al., 2019.

Everywhere in the text and in the figures we have already re-named the newly synthesized GO NPs obtained after amination with hydroxylamine as “haGO-NH2”.

Comment 2:

I believe that the most interesting comparison in term of biological activity should be performed between GO-NH2 NPs (ref. 21) and the newly synthetized NPs presented in the present paper. Authors try to discuss these differences in the discussion section, but it is quite misleading if it is possible to compare just physico-chemical characteristics and not their biological meaning on the same cellular model. Authors scarcely discussed this point (lines 323-327) without providing a convincing justification on the different activity observed in the two cell models.

Answer:  

We agree with the reviewer that although in the discussion we have tried to compare both types of the aminated GO NPs (GO-NH2 versus haGO-NH2), the design of both research works is not alike which makes the comparison of biological activity of both aminated GO difficult. Indeed, in this study we wanted to emphasize on the mitochondrial dysfunction of cells exposed to GO and aminated GO NPs. Regardless these point of view we did follow the recommendation of the reviewer and we have built a comparative table in the Discussion part. It compares not only some of the physico-chemical characteristics of both types of aminated GO nanoparticles but also some important biological characteristics (lines 350-390).

Comment 3:

Why the authors decide to switch from colon cancer to hepatocarcinoma cell model? The cellular mechanisms they analyzed are general, non-cell specific and for this reason they could have used the same cell model to compare the two different GO-NH2 NPs. Please comment.

Answer:  

We have decided to examine the effect of pristine and aminated GO nanoparticles on hepatocellular cancer cells because the hepatocellular carcinoma is one of the most challenging cancer types for studying and for development and design of targeted treatment (see lines 89-99 in the revised text). This is one reason for our choice from one side, while from the other, it is well known that HepG2 is one of the most widely used cell line to study cytotoxicity. Therefore, we have decided that it would be a brilliant model in our experiments.

Comment 4:

It is not possible to appreciate differences in cell morphology according to the images published by the authors, due to poor magnification and the lack of a consistent analysis of morphological markers of cytotoxicity. It is clear that this microscopic analysis is not enough to see any difference.

Answer:  

The micrographs of Neural red stained cells non-treated and treated with GO and haGO-NH2 are with good resolution and quality. Indeed, no significant morphological changes in the GO and haGO-NH2 treated cells were observed. Furthermore, we state that the morphology of almost every individual cell can be seen very clearly. In order to strengthen these observations, we have included micrographs from FDA staining of cells. Please see Figure 4B in the revised version of the manuscript. The resolution is good and allows the reader to see that no significant changes in fluorescent, i.e. viable cells is observed when HepG2 cells are treated with GO and haGO-NH2.

Comment 5:  

More, how the authors justify the different results recorded by WST-1 and LDH assay?

Answer:

Following this remark, we have added to Figure 4 a palette of fluorescent images with FDA stained cells (Figure 4B) treated with both types of nanoparticles. It is easily seen that cells with haGA-NH2 have a reduced number of fluorescent cells, i.e viable cells. The detected, though, concentration non-dependent increased OD with haGO-NH2 probes on Figure 4A is due exactly to the reduced % of viable cells in haGO-NH2 probes. The logic of this experiment in the mode of action of FDA. Only viable cells fluoresce while only cells with compromised or damaged cell membrane can potentially release LDH in the culture medium thus increasing OD.

Comment 6:  

They speculate about the linear regression models they adopted for the two nanoparticles GO and GO-NH2, but probably presenting cell viability results in percentage compared to untreated cells will be more self-explanatory for readers.

Answer:  

We agree that there are different ways to represent the efficiency of the graphenes over the cells’ viability. We have chosen to use regression models because in this way we illustrate the dependence between the OD (a measure for the viability) and the quantities. The model provides the reader with information about the behavior of the OD as a function of the quantity of the graphene. The figure shows inside out the “dynamics” of this dependence. Also from the same picture one can observe that haGO-NH2 saturates at levels of 30 µg/ml and more, a phenomenon which might be explored further.

Comment 7:

Furthermore they could provide IC50 values.”

Answer:  

Following the reviewer’s recommendations, we have already calculated and have provided the IC50 values in Table 1 in the revised manuscript. This table summarizes the physico-chemical characteristics and the biological effects all studied by us types of GO nanoparticles exert on the cells.

Comment 8:

Did Authors collect some data on apoptotic events recorded after GO and GO-NH2 treatment? They listed apoptosis in keywords section, but specific results are missing.

Answer:  

We have already deleted from key words ”apoptosis”.

Comment 9:

Authors discussed LDH literature data (328-342), without providing their conclusion, meaning how their data contribute to clarify contrasting evidence. Please clarify.

Answer:  

We have already discussed this in Comment 5.

Comment 10:

It would be interesting to know ROS kinetic, what happens at shorter time point? In the discussion section Authors discussed about different kinetic according to concentration, but the real kinetic would be the presentation of data at different time points (1, 3, 6 hours, together with 24). Due to hypothesized experimental limits in detection of ROS level, it is possible to ascertain the role of ROS in the cytotoxic effects of GO-NH2 using an indirect method, e.g. pre-treatment with N-acetylcysteine will help to confirm the role of ROS in the cytotoxic effects of GO-NH2.

Answer:  

ROS production was studied in a concentration-dependent manner. The reviewer has right, no time kinetics is represented nor studied by us until now. Could be a good experiment for our next study as we too are interested to learn more about ROS production induced by GO NPs at different time points. ROS kinetics is changed with the word “trends” in the revised version of our manuscript (see line 435 in the revised manuscript).

Comment 11:

I believe that the conclusions on the genotoxic potential of GO-NH2 reported at the end of discussion section are scarcely discussed. Authors declared that data complies with published data on ammonia-modified GO obtained in colon cancer cells, where they assess genotoxicity of these NPs (ref.21). The different characteristics of the nanoparticles could justify a different behavior in term of genotoxicity? Both NPs induced ROS generation, so are differences imputable to the different cell model? Also in this case, the possibility to compare the different GO-NH2 NPs on the same model could had conferred additional value to presented results.

Answer:  

We have added better explanation of his observation. Please see:

“No significant alterations in cell morphology nor substantial DNA damage was detected compared to pristine GO and the control group. In conclusion, ROS production and extracellular acidification rate in mitochondria of GO-NH2 treated cells could be a potential mechanism for cytotoxicity of hydroxylamine modified GO nanomaterials. These findings provide toxicological and mechanistic information that could enrich the knowledge about the molecular mechanisms exerted by pristine and modified GO nanomaterials in different biological systems. Their potential use as anticancer drugs and as vector delivery systems in cancer cells remains to be further elucidated.”

Though further data are needed in order to dissect in details this.

Comment 12:

Authors discussed about “selectivity” reporting this term in the title and abstract section, but I believe this term is misleading. Selectivity would mean activity of NPs assessed in cancer cells versus healthy cells (i.e. non transformed hepatic cells). Here they showed that GO-NH2 NPs had highest cytotoxic activity compared to GO NPs. Please clarify.

Answer:

We have already deleted this word from the title following the other reviewer comment too.

Minor points:

  1. It is not possible to completely read legend of Figure 2- probably there is a problem with pdf. We have corrected Figure 2 where all data now are visible.
  2. Figure 5: why error bars are not present in all columns?” – we have already added all error bars. It was our mistake. Possibly due to file transfer some error bars disappeared. All of this is corrected now.
  3. Positive control for comet assay is missing” - we did not include the positive control here as we prove that the base line outlined by the threshold marking the control cells lack of genotoxicity, i.e, comets, clearly indicates that the genotoxic activity of all GOs at all tested concentrations is very little to completely missing genotoxicity.

Reviewer 2 Report

In this manuscript titled "Amination of graphene oxide leads to increased 2 selectivity and cytotoxicity in hepatocellular 3 carcinoma cells", Georgieva et la have elegantly described exploiting aminated graphene oxide as an anti-cancer therapeutic, rather than simply as a drug delivery agent. The manuscript has been well written and described well. The manuscript would add value to the current set of information. the manuscript may require minor changes.

1) Figure 4 the microscopic pictures of the cells treated Go and Go-NH2.

2)WST-1 Assay needs to be performed in normal liver cells to show that the efficacy is more in cancer cells.Microscopic pictures of the treatment to be displayed. 

Author Response

Reviewer 2:

Submission Date

05 February 2020

Date of this review

13 Feb 2020 21:58:43

Comment 1:

“In this manuscript entitled "Amination of graphene oxide leads to increased selectivity and cytotoxicity in hepatocellular carcinoma cells", Georgieva et al. have elegantly described exploiting aminated graphene oxide as an anti-cancer therapeutic, rather than simply as a drug delivery agent. The manuscript has been well written and described well. The manuscript would add value to the current set of information. The manuscript may require minor changes.”

Answer:

We highly appreciate the positive comments from the reviewer. They very masterly highlight the importance of our work.

Please, find below all comments and responses to the listed reviewer’s comments.

Comment 2: Figure 4 the microscopic pictures of the cells treated GO and GO-NH2.

Answer:

We have followed the reviewer’s recommendation and have added microscopic pictures of fluorescein diacetate (FDA)-stained HepG2 cells treated for 24h with GO and GO-NH2. The general idea of using the FDA method is because of the dye stains only viable cells with the intact plasma membrane. Non-fluorescent FDA molecules which due to disrupted cell membrane pass through are hydrolyzed by intracellular esterases and thus produce fluorescein. The membrane-impermeable fluorescein accumulates in the cytoplasm and exhibits green fluorescence. Fluorescent images in the new Figure4B added in the revised version of the manuscript clearly show that the number of viable cells is severely reduced in GO-NH2 treated cells. The last is a strong result showing that aminated GOs compromised at a greater degree the cell membrane of HepG2 cells than did the pristine ones.

Comment 3: WST-1 Assay needs to be performed in normal liver cells to show that the efficacy is more in cancer cells. Microscopic pictures of the treatment to be displayed. 

Answer:

We do agree with the reviewer that distinguishing between normal and cancer cells' responses to pristine and aminated NPs is an excellent approach in the main aim to study in detail the biological characteristics of the tested nanoparticles. And we plan, to execute this in our future experiments where different cell lines, with different origins, will be used for probing the effect of pristine, aminated and differently modified GOs. We also plan to study the effect of the tested nanoparticles on mitochondria, isolated from rat liver. And we do have some preliminary results which are not ready for publication yet. As a matter of fact, they are part of important experiments planned as research tasks in one of our recently approved projects.

In this particular work our intention was to accumulate, analyse and communicate more data about the biological effects of GO and GO-NH2 nanoparticles, including our data from experiments with HepG2 cells and to compare them with our previous results with Colon 26 cell line (reference 21 in the current work).

We could refer here to some of our previous results with embryonic cells treated with GO and GO-NH2 (ammonia-modified) NPs. These results are published in reference 22 (cited in this work) and they unambiguously demonstrate that no negative effect from the action of the tested GO NPs on normal cells was observed, thus proving a cancer cells’ selective mode of action. Based on these we could suppose that the studied nanoparticles probably would not exert any effect on normal hepatocytes, though this speculation requires more experiments and is not a central point in the current manuscript.

Reviewer 3 Report

Manuscript ID:  ijms-725649

Title:  Amination of graphene oxide leads to increased selectivity and cytotoxicity in hepatocellular carcinoma cells

Authors:  Georgieva, M., et al.

This manuscript examines the toxicity of graphene oxide (GO) and aminated graphene oxide (GO-NH2) particles in hepatocellular carcinoma cells (HepG2).  Toward this end the authors state they have developed a new way to aminate GO that is simpler than other methods and utilizes hydroxylamine.  The GO-NH2 particles were characterized by TEM, DLS, and zeta potential.  Toxicity was assessed by comparison of GO and GO-NH2 in cell viability, morphology, membrane integrity (LDH), ROS production, metabolism (seahorse), and comet assays (over 4-50 ug/mL).  Overall, generally, GO was similar or less toxic than GO-NH2 by a small amount (1-2 fold).

The authors have conducted a range of assays comparing GO with GO-NH2 in HepG2 cells.  The general trends are found throughout the assays that the GO-NH2 nanoparticles exhibit a similar or a greater toxic effect.  The effect is not large.  However, the studies serve as a base-line for studies that utilize these particles for the core of particles that can be further elaborated (e.g., coated, coated with drug, baring targeting molecules, etc.) 

There are concerns, however.  First, it is not clear that the authors claim, that they have prepared aminated GO particles, is correct.  There is no doubt that they have modified the GO particles.  However, the XPS data are not clear in this regard.  The XPS of GO shows a considerable N1s peak while other reports of GO-NH2 do not report this peak (e.g., ‘Efficient amine functionalization of graphene oxide…’, RSC Advances 5(74) DOI: 10.1039/C5RA07892J) and suggest issues with the measurement.  Second, the zeta potential of the GO-NH2 particles prepared by the authors is -12.28 eV while the authors also report the zeta potential for commercially available GO-NH2 is +38.5 eV (see page 12, lines 310-311).  These cannot be the same and experiments need to be done to further characterize the particles the authors have prepared (which are clearly different from GO).

At a minimum, the authors could compare GO with the GO-NH2 prepared by their method and commercially available GO-NH2 (particle characterization and toxicity).  Perhaps the only difference is the degree of amination and this may be reflected in data collected on GO-NH2 nanoparticles for both sources. 

The title says ‘…increased selectivity and cytotoxicity…’.  There are no experiments done that demonstrates selectivity.  The term selectivity should be removed from the title. 

Author Response

Reviewer 3:

Submission Date

05 February 2020

Date of this review

17 Feb 2020 19:39:18

Comment 1: This manuscript examines the toxicity of graphene oxide (GO) and aminated graphene oxide (GO-NH2) particles in hepatocellular carcinoma cells (HepG2).  Toward this end the authors state they have developed a new way to aminate GO that is simpler than other methods and utilizes hydroxylamine.  The GO-NH2 particles were characterized by TEM, DLS, and zeta potential.  Toxicity was assessed by comparison of GO and GO-NH2 in cell viability, morphology, membrane integrity (LDH), ROS production, metabolism (seahorse), and comet assays (over 4-50 ug/mL).  Overall, generally, GO was similar or less toxic than GO-NH2 by a small amount (1-2 fold).

The authors have conducted a range of assays comparing GO with GO-NH2 in HepG2 cells.  The general trends are found throughout the assays that the GO-NH2 nanoparticles exhibit a similar or a greater toxic effect.  The effect is not large.  However, the studies serve as a base-line for studies that utilize these particles for the core of particles that can be further elaborated (e.g., coated, coated with drug, baring targeting molecules, etc.) 

Answer:

We appreciate this meaningful interpretation by the reviewer of our results. Yes, the cytotoxic and genotoxic effects of aminated GO-NH2 in HepG2 cells is not as explicit as initially expected, but the detected differences between pristine and aminated GO NPs are statistically significant in respect to cell viability, proliferation and ROS production. Indeed, these results are planned to serve as a background, as a “base-line”, as the reviewer states, in all further experiments aiming at utilizing these particular type of nanoparticles in the development of therapies for certain diseases.

Comment 2: There are concerns, however.  First, it is not clear that the authors claim, that they have prepared aminated GO particles, is correct.  There is no doubt that they have modified the GO particles.  However, the XPS data are not clear in this regard. The XPS of GO shows a considerable N1s peak while other reports of GO-NH2 do not report this peak (e.g., ‘Efficient amine functionalization of graphene oxide…’, RSC Advances 5(74) DOI: 10.1039/C5RA07892J) and suggest issues with the measurement.”

Answer:  Hydroxylamine amination of GO like all other protocols for modification of GO NPs requires improvement. We do accept the fact that in our case we have obtained only 1.86 % N1, suggesting that we need a higher percentage of -NH2 groups.  However, even though the effect of hydroxylamine amination on the observed physicochemical properties of GO-NH2 is significant in respect to observed parameters like reduction in the size of GO NP thus increasing their permeability into cells; increase in the measured zeta potential, and shrinking and wrinkling of the GO-NH2 sheets.

The article cited by the reviewer shows the C1s and N1s core lines work together with their deconvolution. Very possibly the differences between the results in the work by Navaee et al. and our work derive essentially from the different materials analyzed. Concerning the XPS, we observe that in our and Navaee et al., results the C1s deconvoluted in similar components representing the graphitic C=C bond, the C-N bonds at ~285.8eV and the oxidized carbon bonds at higher binding energy values. As observed by the reviewer some differences are found in the case of the N1s core line. In the Navaee et al., work authors report that bonds with carbon and in particular the component at 398.4eV are assigned to C=N as in our work, while the main peak at 399.4eV is assigned to C-NH2 and the other two components are assigned to C-N-C and CN+ at 400.8eV and 402.1eV, respectively. Moreover, the component at higher BE is assigned to N-O. In our case, the interpretation of N1s was made referring to Beamson G. & Briggs D. “The High Resolution XPS of Organic Polymers – The ESCA 300 Database”, J. Wiley Ed., Chichester 1992 and to other literature (see for example “XPS in development of chemical sensors”, RSC Adv. (2015), 5, 83164, “XPS and NEXAFS studies of aliphatic and aromatic amine species on functionalized surfaces”, Surf. Sci. (2009), 603, 2849). In these works, amine and C=N bonds are found at ~398.5 eV, in the range 399.07 – 399.9 eV fall the amide/imide bonds, the protonated amine C-NH3+ falls at 401.5 eV. At higher BE is found the ionized bond CH3N+ bond while NO2 bonds are at around 405 eV. Our interpretation of the nitrogen spectra was made in agreement with these assignments.

In GO flakes N essentially derives from the exfoliation process in which nitric + sulphuric acids are utilized. The acid treatment leads to strong oxidation of the carbon atoms as it appears from our XPS spectra (see the C1s in figure 1). In this situation, around 50% of carbon atoms are in an oxidized form. It is then likely that the main part of nitrogen bonds turned into oxidized carbon atoms leading to amide/imide bonds corresponding to the main peak falling at ~399.9 eV (see figure 1 N1s spectrum of GO). The component at ~398.5eV was assigned to C-NH2 and C=N bonds while the component at ~401.5eV was assigned to protonated amines. Both of these components are more intense in the functionalized GO. The main difference between our interpretation and that by Navaee et al, derives from the different material analyzed. In the Navaee’s work authors show the XPS results of reduced graphene oxide. Differently from our GO, the XPS of pristine r-GO shows that the intensity of the N1s feature is negligible (see Navaee et al., figure 2). In fact, the process of graphene oxide reduction causes the release of CO, CO2 thus leading to a strong reduction of the amide/imide bonds and of the N content. This can readily explain why in the work of Navaee et al the amide/imide bonds cannot be assigned.

Comment 3: Second, the zeta potential of the GO-NH2 particles prepared by the authors is -12.28 eV while the authors also report the zeta potential for commercially available GO-NH2 is +38.5 eV (see page 12, lines 310-311).  These cannot be the same and experiments need to be done to further characterize the particles the authors have prepared (which are clearly different from GO).

Answer:  We understand the concern expressed by the reviewer. However, at that time we have no reason to doubt in our zeta potential measurements of ammonia-modified GO.  Unfortunately, we can not repeat these measurements because at this time we are run out of commercially available GO-NH2. Moreover, now Sigma offers ammonia-modified GO but even to buy it and to measure again its zeta potential we are not sure that this GO-NH2 is obtained by the same protocol. Indeed, we have been previously asking Sigma about the properties of their ammonia-modified GO but unfortunately, they did not provide any data. The ammonia-modified GO at that time was the reason to aminate GO by ourselves. Moreover, we have checked in the literature about some data for measurements of zeta potential of other ammonia-modified GOs and what we have found was that ZP varies a lot. For example, Cheng Y.W. et al. (2019) have demonstrated very similar positive value of 33.2 mV for zeta potential of aminated by ammonia solution GO nanoplatelets, (https://doi.org/10.1016/j.surfcoat.2020.125441), while the others Verma S. and Dutta R.K.,2015 (DOI: 10.1039/c5ra10555b) have measured the zeta potentials over pH range between 2 and 9 and they found ZP to vary from +3.66 to -25 mV.

All these data have been already discussed in Table 1 in the Discussion section in the revised version of the manuscript. Follow lines 356 to 389 in the text.

Comment 4: At a minimum, the authors could compare GO with the GO-NH2 prepared by their method and commercially available GO-NH2 (particle characterization and toxicity). Perhaps the only difference is the degree of amination and this may be reflected in data collected on GO-NH2 nanoparticles for both sources.

Answer:  

We thank the reviewer for this suggestion as it induced us to summarize our data from previous and recent research on both types of aminated GO NPs and in the revised version of the manuscript, we have prepared a comparative table (Table 1 lines 365-389 in the text) with some of the studied physicochemical and biological characteristics of GO, ammonia-modified GO-NH2 and hydroxylamine modified GO-NH2. We believe that now it is much easier to compare the significance, the effects and the advantages of both amination protocols of graphene oxide nanoparticles and correspondingly to much more freely discuss on their potential development as anticancer therapeutics or drug-carriers.

 Comment 5: The title says ‘…increased selectivity and cytotoxicity…’.  There are no experiments done that demonstrate selectivity.  The term selectivity should be removed from the title. 

Answer:  

Following this recommendation by the reviewer we have already deleted the “selectivity” from the title.

Round 2

Reviewer 1 Report

Authors mainly addressed concerns raised from the first revision of the paper, but still some points need clarification before publication.

Notwithstanding the explanation given by the Authors on the different cell model chosen for the experiments on haGO-NH2, I  still think that it will significantly improve the quality of the manuscript a comparison of the two products (GO-NH2 and haGO-NH2) on the same cell model for at least one cytotoxicity test. Cell proliferation, cell adhesion are not absolute values that it is possible to compare as authors report on Table 1. Each cell line has specific characteristics, i.e. duplication time, metabolic activity, intrinsic antioxidant defense, thus, it is misleading compare the aforementioned biological endpoints on different cell lines based on the arbitrary concentration 50 µg/mL. How did Authors choose this concentration to compare the different compounds?

More, the IC50 reported in Table 1 of GO is quite surprising. How it is possible that GO had IC50 so different in the two cell lines, whereas the GO-NH2 and haGO-NH2 are at least in the same order of magnitude? Are IC50 calculated in both cases based on the same test? Please, comment in the discussion section. Speculation on lines 423-428 could be supported by the result obtained by the Authors on the cytotoxicity test that compare all the products on the same cell model, as I previously suggested.

I agree with Authors that Table 1 is useful to compare NH2-GO and haNH2-GO nanoparticles, but I would limit the comparison on objective data, such as chemico-physical data and IC50. Besides, I appreciated the comments in the discussion section between the activity of GO and haGO-NH2, as I already explained the only comparison useful to be described because on the same cell model.

Concerning missing positive control in comet assay test, it is important to underline that positive control would guarantee that experiments were performed correctly. Positive controls are important as internal control especially when results of tested compounds are negative. More, positive controls are necessary to compare experiments performed in different days. This point needs to be fixed.

Please, better describe Figure 4 in the legend of figure, describing A and B.

In my opinion, it is still not clear what Authors stated on lines 539-541. They discussed about apoptosis but, as they commented in the revised version, data on apoptosis are not showed for haGO-NH2. These data could significantly improve the quality of the manuscript, because results showed integrity of membrane, but also LDH release and as the authors stated this event could be related to the cell death mechanism evoked by haGO-NH2 (457-462). In in vitro system, apoptosis will definitely lead to late apoptosis, meaning necrosis, because there are not cells of our immune system to remove dying cells. Apoptosis experiments will contribute to clarify this point.

More, literature references should be added on lines 457-462.

Author Response

Reviewer 1

Submission Date

05 February 2020

Date of this review

02 Mar 2020 14:34:28

We have tried to address all comments raised by the reviewer.

Please see below:

Comment 1

“Authors mainly addressed concerns raised from the first revision of the paper, but still some points need clarification before publication.

  1. Notwithstanding the explanation given by the Authors on the different cell model chosen for the experiments on haGO-NH2, I still think that it will significantly improve the quality of the manuscript a comparison of the two products (GO-NH2 and haGO-NH2) on the same cell model for at least one cytotoxicity test. Cell proliferation, cell adhesion are not absolute values that it is possible to compare as authors report on Table 1. Each cell line has specific characteristics, i.e. duplication time, metabolic activity, intrinsic antioxidant defense, thus, it is misleading compare the aforementioned biological endpoints on different cell lines based on the arbitrary concentration 50 µg/mL. How did Authors choose this concentration to compare the different compounds?”

Answer:

We do indeed understand the concerns expressed by the reviewer about the differences in the cell models and the comparison that we provide between the two modes of amination of the GO NPs. As a matter of fact, our main idea is not to compare per se the two types of aminated GOs but to rather thoroughly dissect the mechanisms of action of the amination of GOs on the model cells, in our case hepatocellular carcinoma and colon cancer cells (pervious work). The comparison provided between the two modes of amination is done for the discussion part and is a bit preliminary. The same is stated by the reviewer and we do agree with this. Moreover, at that particular moment we cannot perform any experiments with GO-NH2 because:

  1. At that moment we are run out of commercially available GO-NH2.

The high price and the lack of ammonia-modified GO on the market was the reason to aminate GO by ourselves by hydroxylamine. We have checked Sigma and found out that it offers again ammonia-modified GO again.

  1. But even to buy it and to perform cytotoxicity test we are not sure that the newly provided GO-NH2 are obtained by the same protocol and whether their physicochemical properties are the same as of the GO-NH2 used by us in the experiments done for our previous manuscript on Colon 26 cells. Indeed, we have been previously asking Sigma about the properties of their ammonia-modified GO but they did not provide any data.

Therefore, experiments with the comparison of the two products (GO-NH2 and haGO-NH2) on same cell model are scheduled in our new project proposal which we have already successfully obtained by the Bulgarian Science Fund. The main idea is comparison of not only these two modes of amination of GOs but also among modifications like PEGylation and other combinations with anti-tumour therapeutics, etc. It needs good logistics and logic in setting up the new experiments. The work that is planned in the project is big and all these recommendations by the reviewer will be of great help for experiments planning.

The statement by the reviewer that it might be insufficient to compare cell adhesion and suppression in cell proliferation of both cell lines is true but to some extent. It is insufficient but far from misleading as these are the modes of action of the tested GOs no matter of the tested concentrations. Our explanation is that though we have followed the recommendations and have omitted these data from the Table 1, we do believe that these biological processes that are inhibited by our NPs (modified by different modes of amination) are a very good proof for the biological mechanism of action of amination of GOs.

The concentration 50 µg/mL NPs was chosen as the highest used and with the model explicit effect and as a very informative one for the inhibitory power of the GOs. What has to be highlighted here is that this inhibitory effect of NPs on both processes is valid for all tested concentrations of GO, GO-NH­2 and haGO-NH2 NPs.

Comment 2

“More, the IC50 reported in Table 1 of GO is quite surprising. How it is possible that GO had IC50 so different in the two cell lines, whereas the GO-NH2 and haGO-NH2 are at least in the same order of magnitude? Are IC50 calculated in both cases based on the same test? Please, comment in the discussion section. Speculation on lines 423-428 could be supported by the result obtained by the Authors on the cytotoxicity test that compare all the products on the same cell model, as I previously suggested.”

Answer:

IC50 values were calculated by means of GraphPad Prism 7 (GraphPad Software, San Diego, CA, USA) based on the % of cell adhesion at 24 h after exposure of cells to the tested NPs. The difference in the IC50 of GO probably come not only from the different cell types but also form the different tests used for calculating the % of cell adhesion - counting the number of attached cell by an automated cell counter (Countess, Invitrogen) for Colon26 cells and WST-1 assay – for HepG2 cells.

We have added this information in the text - lines 324-337 in the revised manuscript:

It should be keep in mind however the different type of the studied cells as well as the different tests used for calculating the % of cell adhesion on which is based the calculation of IC50. In the case with Colon 26 cells it is based on the counting of the number of attached cells by an automated cell counter (Countess, Invitrogen) while for HepG2 cells it is the assay of WST-1”.

Comment 3

“ I agree with Authors that Table 1 is useful to compare NH2-GO and haNH2-GO nanoparticles, but I would limit the comparison on objective data, such as chemico-physical data and IC50. Besides, I appreciated the comments in the discussion section between the activity of GO and haGO-NH2, as I already explained the only comparison useful to be described because on the same cell model.”

Answer:

We have followed the reviewer’s recommendation and have deleted the data for % of cell adhesion and suppression in cell proliferation form the Table 1 – see Table 1-among lines 365 - 366.

Comment 4

“Concerning missing positive control in comet assay test, it is important to underline that positive control would guarantee that experiments were performed correctly. Positive controls are important as internal control especially when results of tested compounds are negative. More, positive controls are necessary to compare experiments performed in different days. This point needs to be fixed.”

Answer:

Figure 7 already corrected and the positive control for genotoxicity is added already. Please see line: 385. Data for genotoxicity on HepG2 cells treated with 5 mM H2O2 are added in both Results section and in M and M one too.

See line: 290: “HepG2 cells treated with 5mM H2O2 for 30 min at 37 °C were used as a positive control for genotoxicity.”

See line: 527: “HepG2 cells treated with 5mM H2O2 for 30 min at 37 °C were used as a positive control for genotoxicity. After treatment the cells were washed in 1xPBS buffer (2.68 mM KCl, 1.47 mM KH2PO4, 1.37 mM NaCl, 8 mM Na2HPO4), pH 7 and subjected to Comet Assay.”

Comment 5

“Please, better describe Figure 4 in the legend of figure, describing A and B.”

Answer:

We have already corrected it – see between lines190-191.

“ Figure 4. Membrane integrity of HepG2 cells treated for 24 hours with pristine and aminated GO NPs

A: LDH assay ofHepG2 cells after 24 hours of incubation in the presenceof different concentrations of GO andhaGO-NH2 nanoparticles.

B: Fluorescent micrographs of FDA-stained HepG2cells incubated for 24 hours in the presence of GO and haGO-NH2 nanoparticles at different concentrations. Magnification 10x; bar 100 μm.”

Comment 6

“In my opinion, it is still not clear what Authors stated on lines 539-541. They discussed about apoptosis but, as they commented in the revised version, data on apoptosis are not showed for haGO-NH2. These data could significantly improve the quality of the manuscript, because results showed integrity of membrane, but also LDH release and as the authors stated this event could be related to the cell death mechanism evoked by haGO-NH2 (457-462). In in vitro system, apoptosis will definitely lead to late apoptosis, meaning necrosis, because there are not cells of our immune system to remove dying cells. Apoptosis experiments will contribute to clarify this point.”

Answer:

At this point we do not centre our attention on apoptosis as a mechanism of action of our NPs modified by hydroxylamine. In the revised version of the manuscript our discussion on apoptosis is focused on ammonia modified GOs, not on haGO-NH2.We are agreeing with the reviewer that data on apoptosis will contribute to clarify the cell death mechanism evoked by haGO-NH2. We shall probably dissect a potential apoptotic mechanism of action in our next experiments. Up to now the mitochondrial dysfunction is something that is really interesting as a mechanism of action of GO NPs and we consider it as central point in our work.

Novel experiments will shed further light on this as a possible mechanism of action.

Following the reviewer’s thoughts we have corrected the statements in the Discussion part accordingly–see line 438:

“In our previous studies we have shown that ammonia-modified GO NPs have the potential to induce DNA damage and apoptosis in Colon26 cancer cells. Hence, we have analysed here the newly synthesized haGO-NH2 NPs and their ability to destroy DNA in HepG2 cells. We have found that hydroxylamine aminated GO induced only slight DNA damage although caused ROS generation and moreover mitochondrial dysfunction. This suggest that the mechanisms through which haGO-NH2NPs exert their biological activities are not centred in the stability and maintenance of the genome integrity but rather in mitochondrial metabolism and oxidative damage. The last is in contrast to the observed DNA damage in ammonia-modified GO in colon cancer cells.”

Comment 7

“More, literature references should be added on lines 457-462”

Answer:

We have added two more references, confirming that LDH release is a marker of necrotic cell death- see line 416: [45, 46] and lines 741-747:

  1. Chan, F.K-M.;Moriwaki, K.; De Rosa, M.J. Detection of Necrosis by Release of Lactate Dehydrogenase (LDH) Activity. Methods Mol Biol. 2013, 979, 65–70. doi: 10.1007/978-1-62703-290-2_7

46.Helm, K.; Beyreis, M.; Mayr, C.; Ritter, M.;  Jakab, M.; Kiesslich, T.; Plaetzer, K. In Vitro Cell Death Discrimination and Screening Method by Simple and Cost-Effective Viability Analysis. Cell Physiol Biochem 2017;41:1011-1019 DOI: 10.1159/000460910

Reviewer 3 Report

Generally the authors have satisfactorily addressed my concerns.  There are just a few minor edits the authors should address:

  1. 4, Figure 2B, The abbreviation for hydroxylamine aminated GO has been changed in this draft to haGO-NH2.However, in figure 2B, the abbreviation has not been updated from GO-NH2 and needs to be.
  2. 4, line 121, ‘ammine’ should be ‘amine’
  3. 5, lines 130-157 are blank. Is something missing?

Author Response

Reviewer 3:

Submission Date

05 February 2020

Date of this review

02 Mar 2020 15:40:06

Generally, the authors have satisfactorily addressed my concerns.  There are just a few minor edits the authors should address:

  1. 4, Figure 2B, The abbreviation for hydroxylamine aminated GO has been changed in this draft to haGO-NH2.However, in figure 2B, the abbreviation has not been updated from GO-NH2 and needs to be.
  2. 4, line 121, ‘ammine’ should be ‘amine’
  3. 5, lines 130-157 are blank. Is something missing?

We have followed all comments by the reviewer and have addressed them accordingly:

  1. Figure 2B is already corrected and haGO-NH2 is annotated correctly.
  2. “ammines” already corrected to amines

See lines 119-120: “The N1s fit components were assigned to C=N andamine bonds (398.4 eV), amide or imide bonds (399.89 eV) and protonated amines(401.6 eV). XPS survey scans proved that haGO-NH2 contains”

  1. lines 130-157 were blank due to track changes. Now all is corrected and no blank lines are present any more.

We thank the reviewer for all comments in the two rounds of the reviewing process of our work.

Round 3

Reviewer 1 Report

I appreciate Authors responses to my comments, and I can understand the difficulties to perform additional experiments. Part of the story is missing, but I hope they could continue their research and provide further inside in the mechanisms of these products.

Minor suggestion

Line 28-29, rewrite the sentence “One of the directions in this research is the search…” research/search

Line 30, “increased toxicity” compared to what?

Line 37-38, please rewrite the sentence. As such is meaningless and you could propose aminated graphene oxide as a potential anticancer strategy, rather than only a drug delivery system.

Figure 7 was added two times in the manuscript

Figure 7: Positive control is not significantly higher than tested compounds, so caution should be put on showed data.

In general, English revision would significantly ameliorate the understanding of the manuscript.

Author Response

Reviewer 1

Submission Date

05 February 2020

Date of this review

11 Mar 2020 14:00:41

Comments and Suggestions for Authors

I appreciate Authors responses to my comments, and I can understand the difficulties to perform additional experiments. Part of the story is missing, but I hope they could continue their research and provide further inside in the mechanisms of these products.

Thank you for this understanding. We continue our work and believe that in the future we shall fill the whole picture with all needed details regarding the mechanism of action of aminated NPs on tumour cells.

Minor suggestion

Line 28-29, rewrite the sentence “One of the directions in this research is the search…” research/search

“One of the directions in this research is the development of biocompatible therapeutics which selectively target cancer cells.”

Line 30, “increased toxicity” compared to what?

“Here, we show that novel aminated graphene oxide (haGO-NH2) nanoparticles demonstrate increased toxicity toward human hepatocellular cancer cells compared to pristine GO.”

Line 37-38, please rewrite the sentence. As such is meaningless and you could propose aminated graphene oxide as a potential anticancer strategy, rather than only a drug delivery system.

“Intrinsically, our current study provides a new rationale for exploiting aminated graphene oxide as an anti-cancer therapeutic, since it could exert an additive cytotoxic effect on cancer cells in addition to its drug delivery function.”

Figure 7 was added two times in the manuscript

Repetition of Figure 7 is omitted in the revised version of the manuscript.

Figure 7: Positive control is not significantly higher than tested compounds, so caution should be put on showed data.

The reason for this is the used concentration of 5 mM hydrogen peroxide and the treatment time. Though we do believe that the demonstrated comparison among control, positive control and treated with GO NPs probes is statistically significant and shows well the discussed by us lack of genotoxicity of the GO NPs (pristine and aminated) on HepG2 cells.

In general, English revision would significantly ameliorate the understanding of the manuscript.

Thank you for all the comments and revisions. We do believe now that the manuscript is ameliorated significantly.